# Polar amplification of orbital-scale climate variability in the early Eocene greenhouse world

Chris D. Fokkema[1], Tobias Agterhuis[1#], Danielle Gerritsma[1], Myrthe de Goeij[1], Xiaoqing Liu[2], Pauline de Regt[1], Addison Rice[1], Laurens Vennema[1], Claudia Agnini[3], Peter K. Bijl[1], Joost Frieling[4], Matthew Huber[2], Francien Peterse[1] and Appy Sluijs[1]

[1]Department of Earth Sciences, Faculty of Geosciences, Utrecht University, 3584CB Utrecht, The Netherlands.
[2]Earth, Atmospheric, and Planetary Sciences Department, Purdue University, West Lafayette, IN 47907-2051.
[3]Dipartimento di Geoscienze, Universita degli Studi di Padova, I-35131 Padova, Italy.
[4]Department of Earth Sciences, University of Oxford, Oxford, OX1 3AN, United Kingdom.
[#]Now at School of Ocean and Earth Science, National Oceanography Centre Southampton, University of Southampton, Southampton, SO14 3ZH, United Kingdom.

Correspondence to: Chris D. Fokkema (c.d.fokkema@uu.nl)

**Abstract.** Climate variability is typically amplified towards polar regions. The underlying causes, notably albedo and humidity changes, are challenging to accurately quantify with observations or models, hampering projections of future polar amplification. Polar amplification reconstructions from the ice-free early Eocene (~56–48 million years ago) can exclude ice albedo effects, but the required tropical temperature records for resolving timescales shorter than multi-million years are lacking. Here, we reconstruct early Eocene tropical sea surface temperature variability by presenting an up to ~4 kyr-resolution biomarker-based temperature record from Ocean Drilling Program Site 959, located in the tropical Atlantic Ocean. This record shows warming across multiple orbitally paced carbon cycle perturbations, coeval with high-latitude-derived deep-ocean bottom waters, showing that these events represent transient global warming events (hyperthermals). This implies that orbital forcing caused global temperature variability through carbon cycle feedbacks. Importantly, deep-ocean temperature variability was amplified by a factor 1.7–2.3 compared to the tropical surface ocean, corroborating available long-term estimates. This implies that fast atmospheric feedback processes controlled meridional temperature gradients on multi-million year, as well as orbital, timescales during the early Eocene.

Our combined records have several other implications. First, our amplification factor is somewhat larger than the same metric in fully-coupled simulations of the early Eocene (1.1–1.3), suggesting that models slightly underestimate the non-ice related — notably hydrological — feedbacks that cause polar amplification of climate change. Second, even outside the hyperthermals, we find synchronous eccentricity-forced temperature variability in the tropics and deep ocean that represent global mean sea surface temperature variability of up to 0.7 °C, which requires significant variability in atmospheric $pCO_2$. We hypothesize that the responsible carbon cycle feedbacks that are independent of ice, snow and frost-related processes might play an important role in Phanerozoic orbital-scale climate variability throughout geological time, including Pleistocene glacial-interglacial climate variability.

## 1. Introduction

The inverse relationship between Earth's meridional temperature gradient (MTG) and mean surface temperature is a feature of global climate change, often termed polar amplification (PA) (e.g., Masson-Delmotte et al., 2013). Polar amplification — here defined as the ratio of high-latitude (>60º) to low-latitude (<30º) warming — is attributed to various climate feedback mechanisms. These feedback mechanisms can be grouped into two categories: 1) surface-albedo changes, mostly from ice, snow and vegetation and 2) non-surface-albedo feedbacks, including the lapse rate feedback, surface temperature radiative

feedback, longwave cloud and moisture feedbacks, and changes in poleward heat transport (Deconto et al., 1999; Caballero, 2005; Held and Soden, 2006; Pithan and Mauritsen, 2014; Stuecker et al., 2018). Together with the more common focal points of climate sensitivity and global emission trajectories, PA will determine the magnitude of future high-latitude warming and therefore influences the rate of sea level rise through polar ice sheet melt and the dynamics of several carbon cycle feedbacks, such as permafrost thawing (Masson-Delmotte et al., 2013). At present, amplified warming is already observed in the Arctic

(England et al., 2021), but the observational timespan of a few decades is still insufficient to decipher the different contributing feedbacks to PA, or project future PA, as many "slow" feedbacks take place on $10^2$–$10^4$ year timescales (e.g., continental icesheets). Furthermore, climate models produce very different or even contrasting results (e.g., a dominant (Taylor et al., 2013a) versus minor (Stuecker et al., 2018) contribution of the albedo feedback to PA), which cannot be validated due to a lack of observational ground truthing. Additionally, in the presence of icesheets, interaction between the ice-albedo effect and

local influences of the icesheet interfere, masking ice-unrelated processes of PA.

  Reconstructions of PA during ice-free climates in the geological past may, in part, overcome the lack of proper analogues in observations. Earth's most recent ice-free climate state occurred in the early Eocene (56–48 million years ago (Ma)), a time characterized by globally elevated temperatures, exceptionally low MTGs and high atmospheric $CO_2$ concentrations (Cramwinckel et al., 2018; Anagnostou et al., 2020; Gaskell et al., 2022). Ice sheets were insignificant or absent given overall

low deep-ocean benthic foraminifer oxygen isotope ($\delta^{18}O$) values (Zachos et al., 2001). Biotic and geochemical evidence also point to frost-free winters and even subtropical conditions in the Arctic realm (Willard et al., 2019; Sluijs et al., 2020), as well as the Southern Ocean coastal and near-shore regions in Antarctica (Pross et al., 2012; Bijl et al., 2013, 2021). Long-term warmth peaked during the Early Eocene Climatic Optimum (EECO; ~53–49 Ma), when the global mean surface temperature reached values of 10–16 ºC higher than pre-industrial (Inglis et al., 2020). Since ice-related albedo feedbacks did not occur on

fast timescales within the early Eocene, reconstructing PA of climate change during this period may help disentangle the contribution of surface albedo and non-albedo feedbacks.

  Recent work has documented a gradual strengthening of the MTG between the early Eocene and the Oligocene, coeval with the multi-million-year $p$CO_2 decline and global cooling trend (Evans et al., 2018; Cramwinckel et al., 2018; Anagnostou et al., 2020; Gaskell et al., 2022). Much of this work estimated the MTG by comparing tropical sea surface temperature (SST) records

to benthic foraminiferal $\delta^{18}O$-based bottom water temperatures (BWTs). The underlying assumption to this approach is that Eocene BWTs reflect Southern Ocean SSTs (specifically that of subpolar gyres), consistent with climate proxy data

(Cramwinckel et al., 2018; Gaskell et al., 2022) and model simulations (Hollis et al., 2012; Zhang et al., 2022). Indeed, multi-million-year climate trends and its PA have now been demonstrated to dominantly represent long-term variability in atmospheric greenhouse gas concentrations and associated feedbacks (Cramwinckel et al., 2018). This low-resolution work

has shown that in the absence of ice-albedo feedbacks, $10^6$-year-timescale climate change was amplified in the southern high latitudes with a constant and linear PA factor of 1.2–2.2 (Cramwinckel et al., 2018; Gaskell et al., 2022).

Given that different timescales have different associated climate feedback mechanisms (PALAEOSENS Project Members, 2012), the current $10^6$-year reconstructions of past PA (Cramwinckel et al., 2018; Gaskell et al., 2022; Liu et al., 2022) are incomparable to present-day climate change on $10^2$–$10^3$ year timescales. While we acknowledge that the paleoclimate record

cannot approach the current rates of change, records of past orbitally forced climate change may provide the closest possible approximation of PA on shorter ($10^3$–$10^4$-year) timescales.

Numerous carbon cycle perturbations, most, if not all, paced by orbital eccentricity, accentuated early Eocene global warmth (Cramer et al., 2003; Lourens et al., 2005; Westerhold et al., 2018; Lauretano et al., 2018) and present suitable targets for $10^3$–$10^4$-year PA assessment. These events are recognized in the sedimentary record by distinct negative stable carbon isotope

($\delta^{13}C$) excursions (CIEs) (Cramer et al., 2003) and carbonate dissolution in deep-ocean sedimentary environments (Leon-Rodriguez and Dickens, 2010), linked to the release of voluminous $^{13}C$-depleted carbon into the ocean-atmosphere system (Dickens et al., 1997). Several of these events have been shown to represent transient global warming phases, notably the Paleocene-Eocene Thermal Maximum (PETM; ~56 Ma) (Kennett and Stott, 1991; Frieling et al., 2017; Tierney et al., 2022) and Eocene Thermal Maximum 2 (ETM-2; ~54 Ma) (Lourens et al., 2005; Harper et al., 2018). For the PETM, which received

most attention among these so-called 'hyperthermals' (Thomas and Zachos, 2000), reconstructions show rapid global warming of ~5 °C (Frieling et al., 2017; Tierney et al., 2022), ocean acidification, acceleration of the hydrological cycle, and biotic change (Zachos et al., 2005; McInerney and Wing, 2011; Carmichael et al., 2017). Similar changes, yet of smaller magnitude, characterize ETM-2 (Stap et al., 2009, 2010; Sluijs et al., 2009; Gibbs et al., 2012; Harper et al., 2018). Quantification of (global) SST change is, however, challenging for ETM-2 and the subsequent post-PETM hyperthermals due to a lack of decent-

resolution (tropical) SST records. Consequently, only for the PETM, reasonable estimates of millennial-scale PA are available, arriving at 1.7–2.7 (Frieling et al., 2017) and ~1.6 (Tierney et al., 2022). While high-latitude surface warming can be quantified for the subsequent hyperthermals by state-of-the-art benthic $\delta^{18}O$ compilations on timescales of $10^3$-years (Cramwinckel et al., 2018; Gaskell et al., 2022), estimation of PA on this timescale remains impossible as long as the response of the tropical endmember is unknown.

Here, our aim is to solve the lack of high-resolution tropical climate constraints and consequently allow reconstruction of early Eocene $10^4$-year timescale PA. Accordingly, we produced up to ~4-kyr resolution TEX$_{86}$-based SST records over a ~2 Myr early Eocene interval from Ocean Drilling Program (ODP) Leg 159 Site 959 (Fig. 1a; equatorial Atlantic, paleolatitude = ~9 °S (paleolatitude.org version 2.1; van Hinsbergen et al., 2015)), covering multiple early Eocene carbon cycle events. While ideally multiple sites are averaged to track climate variations over a complete latitudinal band, previous work on Site 959

showed that long-term ($10^6$-year) Eocene (Cramwinckel et al., 2018) and short-term ($10^3$-year) PETM (Frieling et al., 2019)

biomarker-based SST records provide a good representation of climate variability in the complete tropical band. Indeed, IPCC-class fully coupled climate models, forced by a range of $pCO_2$ values from 1x to 8x pre-industrial values under early Eocene boundary conditions (*i.e.*, early Eocene paleogeography without ice sheets) within the Deep-Time Model Intercomparison Project (DeepMIP) (Lunt et al., 2021) indicate that SST variability at the location of Site 959 is virtually identical to SST variability in the tropical band (Fig. 1b). Using this new tropical SST dataset, we first test if the covered early Eocene CIEs indeed represent global warming phases and assess how equatorial SSTs varied during a series of suspected early Eocene hyperthermals. Subsequently, we estimate ice-free PA on orbital timescales ($10^4$-years) by comparing equatorial SST variability to a benthic $\delta^{18}$O-based deep-ocean temperature compilation (Westerhold et al., 2020).

## 2. Materials and Methods

### 2.1 Study location, sampling, and approach

Lower Eocene sediments were retrieved in 1995 during ODP Leg 159 at Site 959 Hole D. Site 959 is located in the eastern equatorial Atlantic, ~150 km offshore of Ghana (3°37.656'N; 2°44.149'W), on a northward dipping slope of the Cote d'Ivoire-Ghana Transform Margin at 2091 m water depth (Mascle et al., 1996). The lower Eocene pelagic sediments comprise dominantly diagenetically altered biogenic silica, carbonate and clay, with some organic matter, ranging from soft bioturbated greyish micritic clay-bearing sediments to hard white, carbonate-lean porcellanitic sediments with abundant *in situ* pyritic concretions (Mascle et al., 1996; Wagner, 2002).

Following the existing age model that is based on the identification of the PETM (Frieling et al., 2019) and the Middle Eocene Climatic Optimum (van der Ploeg et al., 2018) as well as biostratigraphy (Cramwinckel et al., 2018), Cores 41R to 38R (800.5–764.2 mbsf) were completely sampled at 2-cm resolution (1143 samples). All samples were freeze dried and cleaned from visible drill-mud. We improve the resolution of the previous age model by evaluating for the presence of suspected early Eocene hyperthermals and orbital cyclicity using bulk rock magnetic susceptibility, various bulk rock geochemical parameters and calcareous nannofossil biostratigraphy. Subsequently, we perform high-resolution TEX$_{86}$ paleothermometry to assess tropical SST variability and broadly assess local ecosystem variability using palynology to exclude an influence of local oceanographic variability on our SST estimates. All analyses were carried out at Utrecht University unless stated otherwise.

### 2.2 Calcareous nannofossils

We analyzed 76 additional samples from Site 959D Cores 41R–35R for calcareous nannofossil assemblages. For this, <1 g was mixed from which an aliquot was analyzed at Padova University, Italy. For optimal constraints, 31 samples were taken from Core 39R, achieving an average sample spacing of 22 cm. Calcareous nannofossil zonation follows Agnini et al., (2014) and ages were calculated utilizing the astronomically calibrated magnetochron ages of Westerhold et al. (2017).

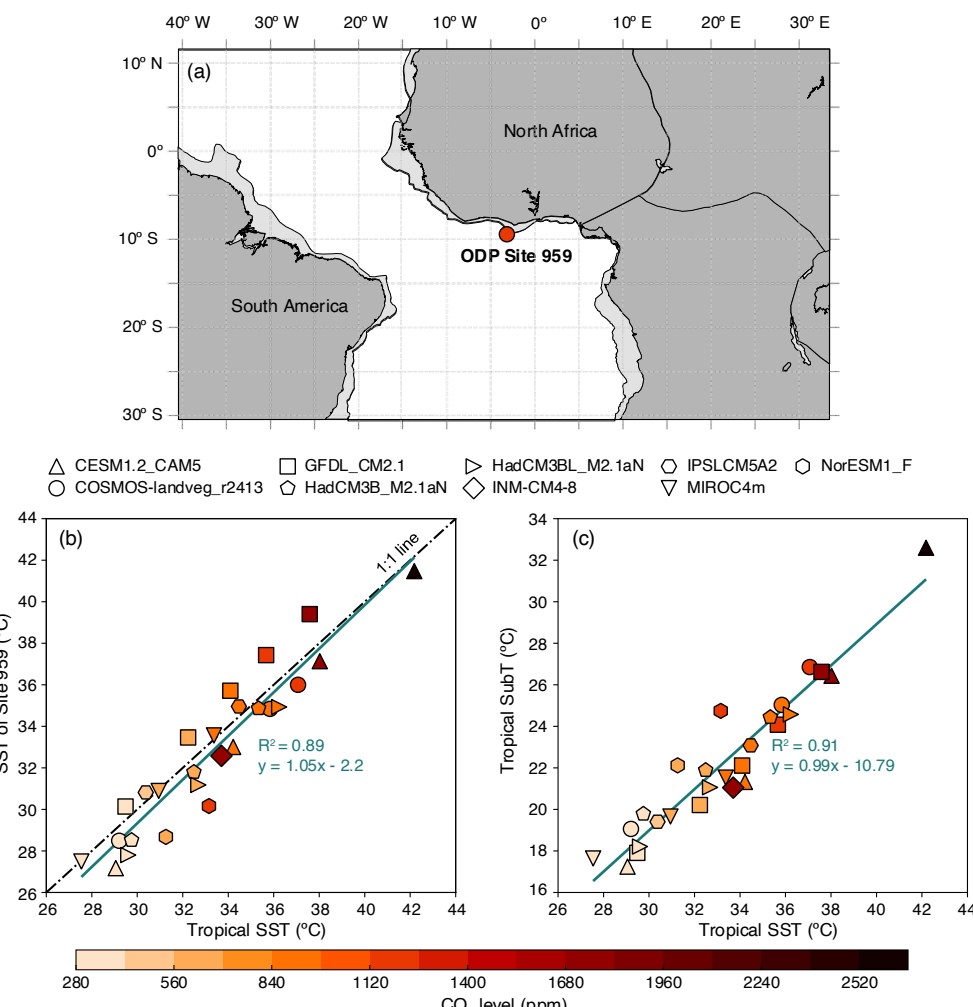

**Figure 1.** Site location and (sub)surface temperature comparisons for Site 959 and the tropical band using the DeepMIP ensemble (Lunt et al., 2021). (**a**) Paleogeographic map depicting the location of ODP Site 959. Light grey areas mark reconstructed continental plates, dark grey areas mark the continents with present-day coastlines. (**b**) Relation between Site 959 SST and low latitude band SST (30ºS–30ºN). (**c**) Tropical SST versus tropical SubT (average of 100–250 m water depth). Shape and color reflect different models and $CO_2$ levels, respectively.

## 2.3 Bulk sediment analysis

Sampled intervals across Cores 41R to 38R were analyzed for bulk sedimentary characteristics to provide, complementary to the calcareous nannofossil biostratigraphy, a chemostratigraphic framework of the early Eocene interval of Site 959. The bulk sediment measurements include magnetic susceptibility (MS), bulk carbonate stable carbon ($\delta^{13}C$) and oxygen ($\delta^{18}O$) isotope ratios, total organic carbon (TOC) content, organic matter stable carbon isotope ratios ($\delta^{13}C_{org}$) and weight percentage calcium carbonate ($CaCO_3$wt%).

### 2.3.1 Magnetic Susceptibility

Bulk MS, a measure for the relative abundance of magnetic (mostly Fe-rich) minerals, was measured on 904 samples. The samples were weighed and put in plastic 40 ml beakers. The measurements were carried out with a MFK1-FA Multifunction Kappabridge. The values of bulk MS are reported in $\chi$. Analytical precision, determined by the standard deviation (SD) of replicate measurements, was better than $5 \times 10^{-10}$ $\chi$.

### 2.3.2 Carbonate oxygen and carbon isotopes

Bulk carbonate isotopes were measured to trace (stratigraphically relevant) $\delta^{13}C$ variations of surface ocean dissolved inorganic carbon (DIC) and potentially trace temperature-forced $\delta^{18}O$ variations of biogenic carbonate. For 828 samples, between 100 and 2200 µg of powdered sediment was analyzed on a Thermo Finnigan GasBench II system, coupled to a Thermo Delta-V mass spectrometer. Isotope values were calibrated to an in-house carbonate standard `NAXOS` and international standard `IAEA-CO-1`. All isotope values are reported against Vienna Pee Dee Belemnite (VPDB). Analytical precision, as determined by the SD of the in-house standard was better than 0.07‰ for $\delta^{18}O$ and 0.07‰ for $\delta^{13}C$. For each sample, the $CaCO_3$wt% was estimated by comparing the signal-to-mass ratio to that of the pure carbonate standards. Precision of this method was better than 13%, based on the SD of the standards.

### 2.3.3 Organic carbon isotopes and content

A selection of 400 samples was analyzed for $\delta^{13}C_{org}$, which in addition to the inorganic $\delta^{13}C$ was used for carbon isotope stratigraphy. For this, first 0.3 g of powdered sample was weighed in a 25 ml plastic Greiner tube and treated with 15 ml of 1 M HCl to remove carbonates. The samples were washed with UHQ and dried in an oven for 62 hours at 60 °C. Approximately 15 mg of dried and homogenized residue was used to determine TOC content with a Fisons CNS analyzer. Bulk total organic carbon isotope ratios were measured with a Finnigan DELTA plus IRMS, coupled to the Fisons elemental analyzer. Isotope values were calibrated against inhouse standards nicotinamide and 'GQ' and reported against VPDB. Precision was determined by the SD of the GQ standard and arrived at better than 0.04‰ $\delta^{13}C$ and 0.07% for TOC content. Sample weights before and after decalcification were compared to provide an additional estimate of $CaCO_3$wt%, following a similar procedure as previous work on the PETM interval of Site 959 (Frieling et al., 2018).

### 2.4 Palynology

To evaluate the influence of local or regional environmental factors (e.g., upwelling, terrestrial input) on temperature variability at Site 959, we analyzed palynological assemblages throughout Core 39R. Specifically, we analyzed fossil dinoflagellate cysts (dinocysts) assemblages, which have proven to be sensitive recorders of Paleogene surface water conditions (Sluijs et al., 2005; Frieling and Sluijs, 2018). For palynological preparation, of 82 samples, 5–10 g of freeze-dried sediment were crushed to ~2 mm chunks. A tablet containing a known amount of *Lycopodium clavatum* spores was added to enable quantitative analysis

of organic microfossils (Stockmarr, 1972). Samples were treated with 10% HCl and decanted to remove carbonates. Next, silicates were removed by repeated treatment with 40% HF and 30% HCl and subsequent decantation after each step. Residues were diluted with tap water and sieved between 250-μm and 15-μm meshes to obtain the required particle size. Palynological residues were mounted on microscope slides by mixing one drop of concentrated, homogenized, residue with glycerin jelly, and covering it with a cover glass. Palynological analysis of dinocysts was performed under 400× magnification until 200 dinocysts were counted, or no material was left.

## 2.5 Lipid biomarkers

### 2.5.1 GDGT analysis

To reconstruct SST variability at Site 959, we applied $TEX_{86}$, a lipid biomarker proxy based on the temperature-regulated homeoviscous adaptation of Nitrososphaeral (previously called "Thaumarchaeota") cell membrane lipids (glycerol dialkyl glycerol tetraethers; GDGTs) (Schouten et al., 2002). For the analysis of GDGTs, between 3 and 45 g of sediment of 268 samples was powdered and weighed in glass tubes. Due to low GDGT concentrations in some intervals (<0.1 ng/g), some neighboring samples were pooled up to a maximum of 125 g of sediment. Lipids were extracted in a 25 ml solvent mixture of dichloromethane (DCM):methanol (MeOH) (9:1 by volume:volume)) by a Milestone Ethos X Microwave Extraction System, set to 70 °C for 50 minutes, after which 99 ng of a $C_{46}$ GTGT standard was added to enable quantitative analysis. Lipid extracts were filtered over a $NaSO_4$ columns and dried under a $N_2$ blower. The dry lipid extracts were separated in apolar, neutral and polar fractions through $AlO_x$ column chromatography, with hexane/DCM (9:1), hexane:DCM (1:1) and 1:1 DCM:MeOH (1:1), respectively as mobile phases. Fractions were again dried under a $N_2$ blower and weighed. Polar fractions were diluted in hexane/isopropanol (99:1) to a concentration of 2 mg/ml, and pressed through a 0.45 μm polytetrafluoroethylene filter into a 1 ml glass vial. Per sample, 10 μl filtered polar fraction was analyzed by an Agilent 1290 infinity ultra high-performance liquid chromatography (UHPLC) coupled to an Agilent 6135 single quadrupole mass spectrometer with settings according to (Hopmans et al., 2016). Measurements were considered below proper detection limits for application of $TEX_{86}$ when one or more isoprenoid GDGT peaks did not exceed three times background noise (*i.e.*, peak areas below ~2000 units). This resulted in exclusion of 47 samples.

Analytical precision was estimated by analysis of a systematically injected in-house GDGT standard, which resulted in a SD of 0.006 $TEX_{86}$ units, corresponding with 0.2 ºC (using the $TEX_{86}^H$ calibration (Kim et al., 2010)) in the $TEX_{86}$ range of early Eocene Site 959. This error represents the analytical error of temperature variability estimates, which is relevant for PA calculations, so any recorded variability with larger magnitude implies a climatological signal. Robustness of applying this standard-based analytical uncertainty to our dataset was confirmed by repeated measurements (n = 9) of one sample, which resulted in a SD of 0.12 ºC ($TEX_{86}^H$).

### 2.5.2 Indices for non-thermal effects

Multiple GDGT-ratios were examined to assess confounding factors on the $TEX_{86}$, by testing for contribution of GDGTs from terrestrial sources (Hopmans et al., 2004), methanogens (Blaga et al., 2009), anaerobic methane oxidizers (Weijers et al., 2011), methanotrophs (Zhang et al., 2011) and deep-water communities (Taylor et al., 2013b), and check for non-thermal influence on the crenarchaeol isomer (O'Brien et al., 2017) and other non-thermal factors (Zhang et al., 2016) using the R-script by Bijl et al. (2021) (Fig. S1). Two samples were left out of further analysis based on their GDGT-2/GDGT-3 ratio, a measure for deep water contribution, exceeding threshold values of 5 (Taylor et al., 2013b; Rattanasriampaipong et al., 2022). Furthermore, we left out four samples based on a deviation of the expected Ring Index (RI) values relative to their $TEX_{86}$ value ($\Delta RI$) (Zhang et al., 2016), which are positioned throughout the studied interval, but exhibit no apparent pattern with other geochemical records. All other samples are well within the ranges of the modern calibration dataset and are therefore deemed suitable for temperature reconstruction, resulting in a final dataset of 216 datapoints with near-continuous stratigraphic coverage across Core 39R.

### 2.6 $TEX_{86}$ calibration

The calibration of $TEX_{86}$ to temperature has remained challenging since the proxy was first proposed (Schouten et al., 2002; Hollis et al., 2019). Paleotemperature reconstructions obtained by extrapolation of coupled satellite measurements and surface sediment-derived $TEX_{86}$ ratios are dependent on the chosen calibration-model, particularly outside the range of modern ocean temperatures as is the case in many Eocene studies, including this study. The calibration-model choices can be summarized in two paths: linear versus non-linear models, and ocean surface versus subsurface calibrations. The considerations that went into choosing the calibration(s) for this work are described in depth below.

### 2.6.1 Calibration shape

Following the original linear $TEX_{86}$-SST calibration (Schouten et al., 2002), subsequently proposed calibrations include linear models (O'Brien et al., 2017), including a spatially varying Bayesian approach ('BAYSPAR') (Tierney and Tingley, 2014), and as well reciprocal (Liu et al., 2009), and exponential (Kim et al., 2010) models. Linear calibrations are typically preferred because they are the simplest models. However, $TEX_{86}$ describes only a limited component of the response of Nitrososphaeral membrane-lipids to temperature (Schouten et al., 2002). For instance, GDGT-0 and crenarchaeol (cren) are not incorporated in $TEX_{86}$ due to their high variability and to other sources of GDGT-0, including methanogenic archaea (Schouten et al., 2002). Nevertheless, these compounds dominate the isoGDGT temperature response at temperatures above 15 ºC (Kim et al., 2010). If Nitrososphaerales increasingly adapt their membranes using GDGTs that are not included in $TEX_{86}$ at higher temperatures, it follows that $TEX_{86}$ might lose sensitivity to temperature at the higher temperature range (Cramwinckel et al., 2018). This variable degree of (in)sensitivity presumably results in a non-linear relationship over the complete temperature range.

The BAYSPAR calibration partly accounts for variable $TEX_{86}$-SST relationships by generating linear regressions from selected analog locations from the surface sediment dataset based on given $TEX_{86}$ search tolerances (Tierney and Tingley, 2014). This approach is, however, problematic for datasets that have $TEX_{86}$ values far beyond the surface sediment-derived

$TEX_{86}$ values, including that from the Eocene of Site 959. Using BAYSPAR for Eocene Site 959 $TEX_{86}$ data requires a large search tolerance and results in extrapolation of a constant linear slope based on a rather small number of warmest analog locations. Another linear calibration (O'Brien et al., 2017) uses a linear regression between the global surface sediment $TEX_{86}$ database and SST for regions warmer than 15 °C, resulting in a calibration that has an approximately similar slope to the BAYSPAR calibration in the high $TEX_{86}$ range.

The exponential $TEX_{86}^{H}$-SST calibration (Kim et al., 2010) (which excludes (sub)polar and Red Sea data, which have anomalous GDGT distributions) presents a relatively good fit with the non-linear behavior between fractional GDGT abundances and SSTs, and has therefore often been applied in climate reconstructions of past warm intervals (Cramwinckel et al., 2018; Frieling et al., 2019). However, significant drawbacks of the $TEX_{86}^{H}$-calibration are regression dilution and the structured residual errors with respect to modern core-top dataset (Tierney and Tingley, 2014).

The modern surface sediment dataset shows that $TEX_{86}$ has very little sensitivity to temperature variability below 15 °C, indicating non-linearity at the low temperature end (Kim et al., 2010). The question is whether the contributions of GDGTs included in $TEX_{86}$ keeps increasing beyond the modern surface sediment dataset, or if this response saturates, as argued previously (Cramwinckel et al., 2018). Importantly, the fractional abundance of GDGT-1 versus GDGT-2+GDGT-3+cren' (the constituents of $TEX_{86}$) in the global surface sediment dataset, for samples from which these are available (Tierney and

Tingley, 2015), shows that the $TEX_{86}$-SST relationship is nonlinear (Fig. 2). Specifically, at low SSTs, both GDGT-1 and GDGT-2+GDGT-3+cren' increase towards ~15 ºC. At higher SSTs, GDGT-2+GDGT-3+cren' continues to increase while GDGT-1 reaches a local maximum between ~15 ºC and 20 ºC and decreases at higher temperatures. In the $TEX_{86}$ ratio this leads to minimal sensitivity at low temperatures and high sensitivity at higher temperatures. Because of the non-linearity in GDGT-1 response, linear $TEX_{86}$-SST calibrations are more likely to overestimate SSTs in the high $TEX_{86}$ regime of the surface

sediment dataset (such as the linear model from O'Brien et al. (2017) in Fig. 2a, 2c). In contrast, an exponential calibration model presents a better fit between the high-end surface sediment $TEX_{86}$ and satellite SST data ($TEX_{86}^{H}$ from Kim et al. (2010) in Fig. 2b, 2d). The exponential shape can account for the diminishing sensitivity of $TEX_{86}$ to SST at higher temperatures and suggests that the response of linear calibrations overestimates climate variability in reconstructions of warm climates. It can therefore be argued that extrapolation of this exponential calibration likely yields a more accurate reconstruction of temperature

variability in the early Eocene tropics and we therefore follow the previous inferences as outlined above and in Cramwinckel et al. (2018) about the non-linear relationship between GDGT distributions and temperature, by using exponential calibration models.

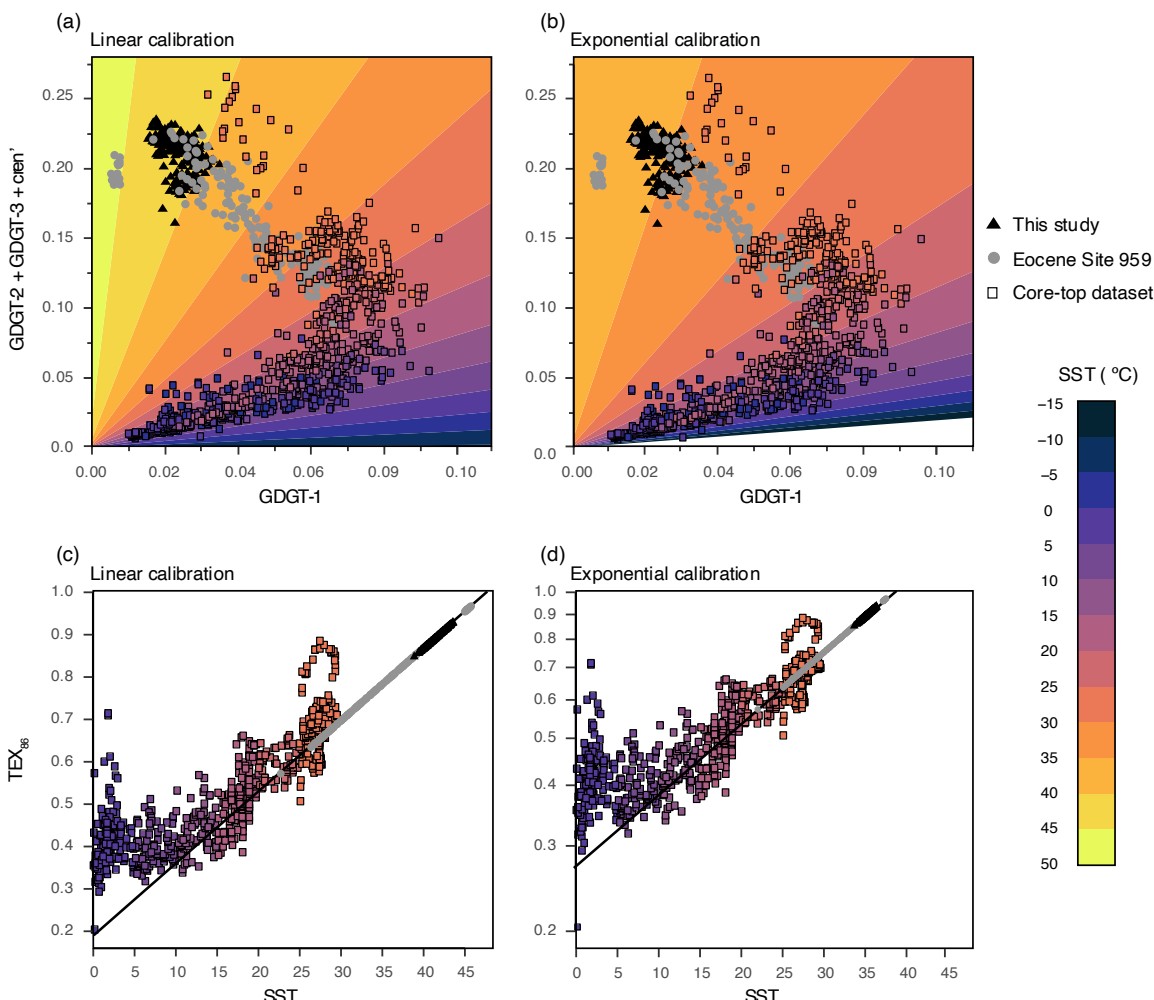

**Figure 2.** Fractional GDGT abundances (**a–b**) and SST-TEX$_{86}$ relationships (**c–d**) for the global core-top dataset and Eocene of Site 959.
(**a–b**) Fractional abundance of GDGT-1 versus GDGT-2+GDGT-3+cren' (i.e. the constituents of TEX$_{86}$) for samples of the global core-top dataset for which these are available (Tierney and Tingley, 2015) and Site 959 Eocene data (Cramwinckel et al., 2018; Frieling et al., 2019; This Study). Color infill of the squares represents World Ocean Atlas 2009 (Locarnini et al., 2010) SSTs. Background colors represent calibrated SSTs based on the linear calibration by O'Brien et al. (2017) (**a**) or TEX$_{86}$$^{H}$ exponential calibration by Kim et al. (2010) (**b**). (**c–d**) Core-top TEX$_{86}$ values and associated SSTs as presented by Tierney and Tingley (2015). The TEX$_{86}$ range of Eocene Site 959 is illustrated by plotting on top of the linear (**c**) or exponential (**d**) calibration model, both represented by black lines.

### 2.6.2 Calibration target depth

Originally, TEX$_{86}$ was calibrated to SST and the temperature at 100 meters depth (Schouten et al., 2002). However the currently accepted view is that that pelagic Nitrososphaerales from below the mixed layer are the dominant source to sedimentary GDGT assemblages (Schouten et al., 2002; Kim et al., 2012; Ho and Laepple, 2016; Tierney et al., 2017; Hurley et al., 2018; van der Weijst et al., 2022). Observations show that cell counts of ammonia oxidizing Nitrososphaerales and GDGT abundances peak

at the base of the NO$_2^-$ maximum, generally positioned between 50 and 100 m in present-day tropical Atlantic Ocean (Zakem et al., 2018). In contrast, the upper 50 m, including the mixed layer, contain few GDGT-producers (Massana et al., 2000; Karner et al., 2001; Sinninghe Damsté et al., 2002; Hurley et al., 2018), because the producers are relatively sensitive to photoinhibition and generally outcompeted (Merbt et al., 2012).

The integrated source depths of GDGTs from sediment samples can be estimated using the GDGT-2/GDGT-3 ratio, which shows a correlation with water depth in the core-top dataset (Taylor et al., 2013b; van der Weijst et al., 2022; Rattanasriampaipong et al., 2022). More importantly, this depth-dependence seems to stem from the depth in the water column were GDGTs are produced (Hernández-Sánchez et al., 2014; Villanueva et al., 2015; Hurley et al., 2018). These studies show that there is a dramatic shift in GDGT-2/GDGT-3 values at a water depth of approximately 200 m, after which GDGT-2/GDGT-3 values in suspended particulate matter rapidly increase to values >5 with increasing depth (Hurley et al., 2018). The GDGT-2/GDGT-3 values for the studied interval of Site 959 are generally below 4 (Cramwinckel et al., 2018; Fig. S1) and, considering integrated depth-GDGT-2/GDGT-3 relationships (van der Weijst et al., 2022), we therefore argue for a dominantly shallow (<200 m) source of the GDGTs. Based on the GDGT-2/GDGT-3 ratios at this site, and given generally low concentrations of GDGTs shallower than 50 m in the modern open ocean (Hurley et al., 2018), we infer that peak integrated GDGT source depth is likely between 50 and 200 m water depth for the early Eocene at Site 959.

While 50–200 m is still relatively shallow water, it includes upper thermocline waters in many locations, including Site 959 (van der Weijst et al., 2022). Therefore, much of the present-day surface sediment GDGT distributions and those from Site 959 likely dominantly comprise GDGTs that originate from below the mixed layer. Calibrating core-top TEX$_{86}$ data to SST might therefore lead to an unrealistically shallow temperature-TEX$_{86}$ slope when extrapolating to Eocene temperatures, because the meridional temperature gradient decreases with water depth (Ho and Laepple, 2016).

The existing subsurface temperature (SubT) TEX$_{86}$ calibrations target different integrated depth ranges and have different calibration model choices. Based on exponential calibration models, Ho and Laepple, (2016) proposed an ensemble of depth-integrated TEX$_{86}$-temperature calibrations up to 1000 m depth and with TEX$_{86}$ both as a dependent and as an independent variable. Based on above inferences about peak GDGT source depths at early Eocene Site 959, we choose an equally weighted depth range from this ensemble that targets the interval between 100 and 250 m. This calibration, to which we refer as "SubT$_{100-250m}$", gives an estimate of shallow subsurface temperature variability which is close to our expected GDGT sourcing depths. The SubT calibration should be considered a conservative estimate of temperature response (relatively low sensitivity with a given TEX$_{86}$ change) as it adds 50 m to the inferred GDGT source depths, integrating seawater temperatures down to 250 m. Another exponential SubT calibration that focuses on the relevant depth range is published by Kim et al. (2012) (here termed 'SubT$_{Kim2012}$'), which is calibrated to the upper 200 m water depth and thus includes the mixed layer, which increases proxy sensitivity.

### 2.6.3 Calibration choice for polar amplification assessment

As the PA factor is independent of reconstructed absolute temperatures, only the calibration slopes are relevant for the estimation of PA factors. Importantly, as climate models show that variability of SST and SubT is equal (Fig. 1c), consistent with data-based estimates (Ho and Laepple, 2016), this allows the in-tandem use of SST and SubT calibrations to calculate PA factors and provide an estimated error range of SST variability. By covering a relatively large range of water depths, this calibration approach furthermore accounts for possible uncertainties in the early Eocene water column structures, including

mixed layer and nitracline depths. Three exponential calibration models, $TEX_{86}^{H}$, $SubT_{100-250m}$ and $SubT_{Kim2012}$ which we, based on above argumentation (see section 2.6.1, 2.6.2) argue to present most realistic slopes in the high temperature end of the Eocene tropics, are plotted in Fig. S2. The calibration slopes of these three calibration models at the high temperature end are, relatively steep ($TEX_{86}^{H}$), relatively shallow ($SubT_{100-250m}$) and intermediate ($SubT_{Kim2012}$). To provide a conservative error estimate regarding SST variability, we converted our $TEX_{86}$ record to temperature using calibrations for both SST ($TEX_{86}^{H}$)

(Kim et al., 2010) and the shallow subsurface ($SubT_{100-250m}$)) of Ho and Laepple (2016), that together cover the plausible range of $TEX_{86}$-SST response. In summary, we consider the range between $TEX_{86}^{H}$, the 0 m surface ocean end-member, and the $SubT_{100-250m}$ to cover an appropriate range of possible SST variability, and this range is used as error range for calculation of PA factors.

### 2.7 Polar amplification assessment

We used the ratio of temperature variability in the tropical surface ocean compared to deep ocean BWTs to assess PA, following the approach of previous work (Cramwinckel et al., 2018). The BWTs are derived from the benthic oxygen isotope data as provided in the "CENOGRID" compilation by Westerhold et al. (2020). We followed the recommendations of Hollis et al. (2019) for the $\delta^{18}O$-temperature conversion, by applying the equation of Kim and O'Neil (1997) as modified by Bemis et al. (1998), an ice-free $\delta^{18}O_{sw}$ of -1.0‰ (Standard Mean Ocean Water; SMOW) and a -0.27‰ conversion factor from SMOW

to VPDB (Hut, 1987). We assume a constant analytical error of 0.36 ºC (0.08‰), based on the maximum published error of the $\delta^{18}O$ data included in the CENOGRID between 54 and 52 Ma (i.e. Littler et al., 2014; Lauretano et al., 2015, 2018; Thomas et al., 2018). Absolute $\delta^{18}O$-based BWT reconstructions are currently challenged by recent advances in carbonate clumped isotope thermometry (Meckler et al., 2022). However, on shorter timescales, clumped isotope data support the magnitude of early Eocene BWT variability from $\delta^{18}O$-based estimates (Agterhuis et al., 2022). The long timespan, high resolution, and

combination of multiple locations of the CENOGRID makes this record most appropriate for reconstructing BWT variability for our study. It should be noted that the amplitude of short-term variability might be slightly dampened compared to single-site BWT records (Fig. S3). The dampened variability may imply that comparison to the CENOGRID gives a conservative estimate of PA.

Calculation of (orbital-scale) PA by comparing tropical $TEX_{86}$-derived SST variability from Site 959 to the benthic $\delta^{18}O$-
derived BWT variability from the CENOGRID compilation (Westerhold et al., 2020) relies on multiple underlying

assumptions. Principally, we assume that the variability captured in the $TEX_{86}$ signal retrieved from sedimentary sequences of Site 959 represents the SST variability of the complete tropical band. This assumption is justified by the closely related SST variability at Site 959 and the whole tropical band in data (Frieling et al. 2019) and within the DeepMIP climate model ensemble (Fig. 1). Moreover, we find no evidence for changes in local environments that might influence GDGT distributions

(Fig. S1) and also our palynological associations are indicative of stable open marine conditions throughout the studied interval (Fig. S4). Further, we assume (post)depositional processes (e.g. bioturbation) have not reduced the variability of the record on the studied time scale (>~9 cm; 20 kyr). For the deep ocean temperature signal, we assume that the amplitude of variability equals that of the high-latitude Southern Ocean, likely the dominant locus of deep-water formation throughout the early Eocene (Hollis et al., 2012; Huck et al., 2017; Zhang et al., 2022). Relatively stable deep-water formation throughout the early Eocene

is suggested by general consistency between benthic foraminifer $\delta^{18}O$ and $\delta^{13}C$ records of the Atlantic and Pacific ocean basins (Westerhold et al., 2018). Furthermore, deep-water formation within the DeepMIP model ensemble is relatively insensitive for $pCO_2$ changes in the range of early Eocene hyperthermals and paleogeography (Zhang et al., 2022).

Polar amplification was calculated by three different methods, which increasingly rely on stratigraphic correlation. As a crude first order approach of comparing short-term variability, the SDs of the (1-Myr LOESS) detrended records were compared.

For the second approach we compared the magnitudes of correlated warming events by a Deming-regression analysis, which included propagated analytical errors of both paleotemperature records to calculate PA for both the SST and SubT datasets. The reported errors of the PA factors represent the standard errors of the associated Deming regression slopes. For comparison with the PETM data published estimates of tropical surface warming (Frieling et al., 2017, 2019) and bottom water warming (Dunkley Jones et al., 2013) were used.

Third, we calculated PA by directly comparing the SST and BWT records in time bins. First, the optimal binning interval was determined based on the criteria that it records climate variability in the 100-kyr-eccentricity band and includes the maximum number of bins with three or more datapoints. The optimum bin size, based on the highest number of bins and included datapoints, lies close to 20 kyr (Fig. S5), which is above Nyquist frequency for short-term (100-kyr) eccentricity. Therefore, a bin size of 20-kyr was applied for the dataset, resulting in 31 bins for Site 959, after excluding all bins that include less than 3

data points. Next, a Deming regression analysis was performed between both binned datasets, incorporating the SD resulted from binning. Note that different bin sizes around the optimum do not result in large offsets in calculated PA (Fig. S5c–d). The same approach was followed for the binning of the long-term datasets in Fig 6a, but with a bin size of 1 Myr. For all three methods of PA calculation, the range between SST- and SubT-based PA was used as estimate of final error range, to cover calibration uncertainty.

Modelled-PA was calculated based on a selection of model runs from the DeepMIP ensemble, from which the output data was retrieved from (Lunt et al. (2021): COSMOS-landveg_r2413 (COSMOS), GFDL_CM2.1 (GFDL), HadCM3B_M2.1aN (HadCM3), CESM1.2_CAM5 (CESM) (Zhu et al., 2019) and IPSLCM5A2 (IPSL) (Zhang et al., 2020). From the output data, spatially weighted, annually averaged SSTs between 30 ºN and 30 ºS was used as tropical SST, and averaged winter SST data for <60 ºS as Southern Ocean winter SST. Precision was determined by the SD of SST data within the selected latitude bands.

## 3. Results

### 3.1 Refined stratigraphic framework

Previous work constructed a low-resolution age-depth model for the Eocene interval of Site 959 using $^{187}Os/^{188}Os$, $\delta^{13}C$ and calcareous nannofossils, supported by the identification of Milankovitch cyclicity in sediment color (Cramwinckel et al., 2018). We here amended the age model for the lower Eocene interval based on higher resolution calcareous nannofossil biostratigraphy. We pair the new biostratigraphic constraints with bulk organic carbon and bulk carbonate $\delta^{13}C$ stratigraphy in a combined age model. This is subsequently used to correlate to global exogenic trends recorded in deep ocean benthic foraminifer isotope stratigraphy (Westerhold et al., 2020).

Our calcareous nannofossil biostratigraphic analysis indicates the same position for the Base of *Tribrachiatus orthostylus* and Top of *T. contortus* in the top interval of Core 40R at 785.05(±0.38) mbsf. Despite the rare presence and poor preservation of calcareous nannofossils in the interval of ~777–778 mbsf, the base of *Discoaster lodoensis* is recorded at 777.84(±0.12) mbsf, previously found to occur just before the K event (Agnini et al., 2009). The Top of *T. orthostylus*, generally found to occur close to the R event (Westerhold et al., 2017), is positioned in the ~9 m core gap above Core 39R.

The calcareous nannofossil biostratigraphy allows correlation of the $\delta^{13}C$ records to long-term trends and short-term carbon cycle perturbations captured in the orbitally tuned benthic foraminifer isotope records (Westerhold et al., 2020), which reflect global exogenic carbon cycle trends (Fig. 3). A gradual decline in bulk carbonate $\delta^{13}C$ between 800 and 784 mbsf (Cores 41R, 40R) corresponds to calcareous nannofossil assemblage zones of CNE2 up to the boundary with CNE3 (~55.6–54.3 Ma), following global deep ocean isotope trends (Westerhold et al., 2020). The small stratigraphic interval recovered by Core 38R (765.09–764.25 mbsf), positioned within Zone CNE5 (~50.6–48.9 Ma), again displays higher $\delta^{13}C$ values, which is also consistent with the long-term trend captured in the benthic carbon isotopes.

Several negative CIEs are recorded in both carbonate and bulk organic $\delta^{13}C$ records in Core 39R (Fig. 3, Fig. 4a). These CIEs typically coincide with drops in $CaCO_3$wt% and peaks in magnetic susceptibility, consistent with other deep ocean records of early Eocene events. However, the very low bulk carbonate $\delta^{18}O$ and $\delta^{13}C$ values ($\delta^{13}C$ down to -6.12‰), strong correlation between $\delta^{18}O$ and $\delta^{13}C$, and overall low $CaCO_3$wt% (Fig. S6) suggest an authigenic carbonate component in the sediment of 39R–38R, consistent with porewater analyses (Mascle et al., 1996; Zachos et al., 2010; Leon-Rodriguez and Dickens, 2010; Slotnick et al., 2012), which compromises reconstruction of absolute $\delta^{18}O$ and $\delta^{13}C$ values and the degree of variability of surface DIC variations. It is therefore unlikely that the large magnitude of recorded bulk carbonate $\delta^{13}C$ excursions (here in the order of 2–4‰) represents $\delta^{13}C$-DIC variability in the surface ocean. The congruence between the major $\delta^{13}C_{org}$ and carbonate $\delta^{13}C$ excursions implies, however, that the stratigraphic positions of CIEs are still recorded in the isotopic signal of the carbonate. Due to the diagenetic overprinting the bulk carbonate $\delta^{18}O$ record is not further considered as a climate signal.

Based on the biostratigraphic boundary of CNE3-CNE4 (~52.9 Ma) at ~777.8 mbsf, the negative CIE at 777.5 mbsf can be identified as the K event (also called 'ETM-3' or 'X'). As the CNE2-CNE3 boundary in the top of Core 40R just predates the H1 ('ETM-2') and H2 CIE events, these events are likely to be positioned in the ~2.6 m core gap between Core 40R and 39R.

The identification of other carbon cycle events can subsequently be inferred using the positions of the CIEs and interpolation between established bio- and chemostratigraphy. Our combined stratigraphic constraints (Fig. 3) show that the recorded negative CIEs at Site 959 represent at least five globally recognized CIEs: I1, I2, J, K and L. Note that we correlate two excursions related to L, here termed L1 and L2, while these events were previously not separated (Westerhold et al., 2018; Lauretano et al., 2018). The here constructed refined age model, including the biostratigraphic and carbon isotope stratigraphic tie-points, is available on Zenodo (https://doi.org/10.5281/zenodo.8309643).

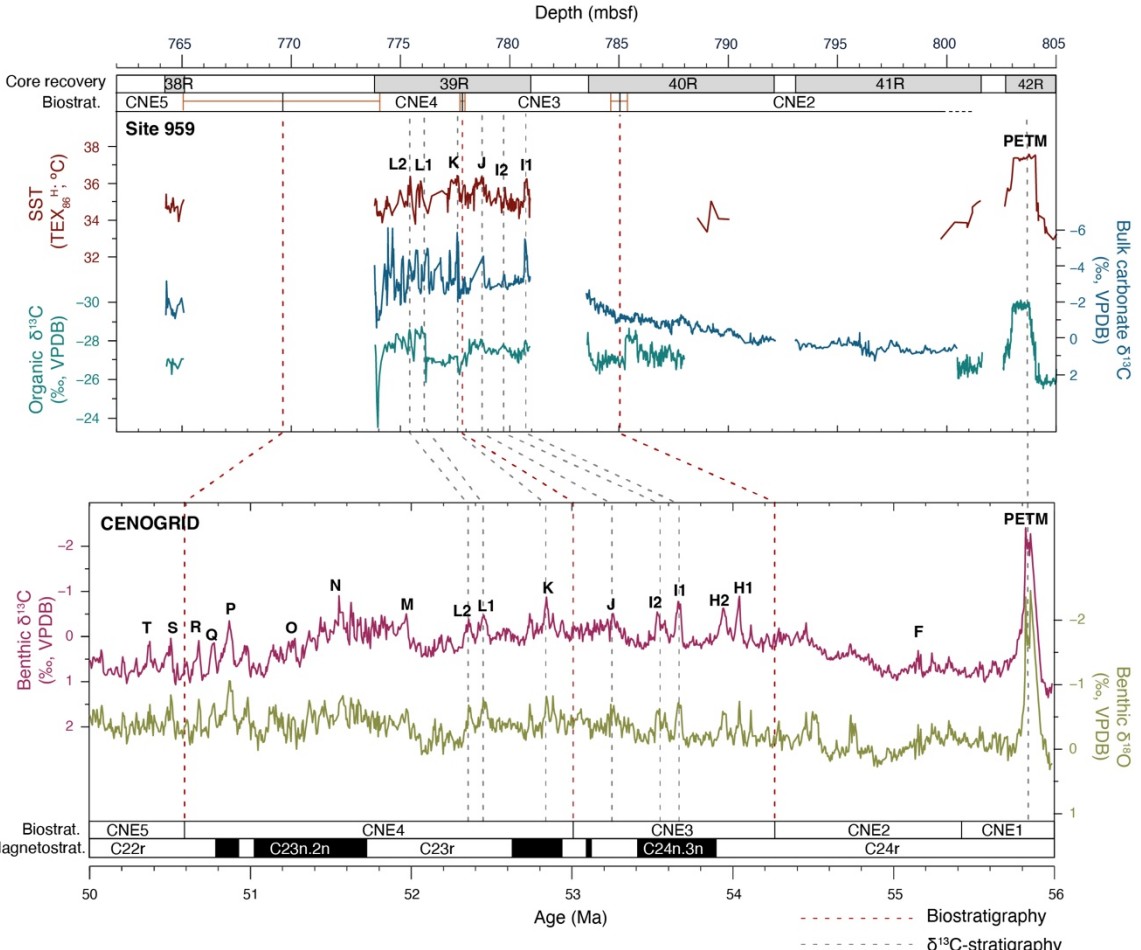

**Figure 3.** Stratigraphic correlation between Site 959 Cores 38R–42R and the CENOGRID benthic compilation. Top of figure: Site 959 data (Cramwinckel et al., 2018; Frieling et al., 2019; This study) with core recovery, nannofossil biostratigraphic zonation (with associated depth uncertainty), TEX$_{86}$, bulk organic δ¹³C and bulk carbonate δ¹³C against depth. Bottom part of figure: benthic δ¹³C and δ¹⁸O from the CENOGRID (Westerhold et al., 2020) and magnetostratigraphic and calcareous nannofossil biostratigraphic zonation (Westerhold et al., 2017). Grey dashed lines indicate correlations based on carbon isotope ratios; red dashed lines indicate biostratigraphic correlations.

Our new TEX₈₆-based SST record, with a median resolution of ~4 kyr across Core 39R, indicates average values of 35.2±0.6 ºC following the TEX₈₆$^H$ calibration (present-day annual average SST is 27.7 ºC (Locarnini et al., 2018). A ~0.7 ºC long-term warming marks the lower part of the record up to ~777 mbsf, after which values drop by ~1 ºC towards ~774 mbsf (Fig. 4). Absolute SSTs derived from one single proxy, including TEX₈₆, should be taken with care, but our new SSTs match the

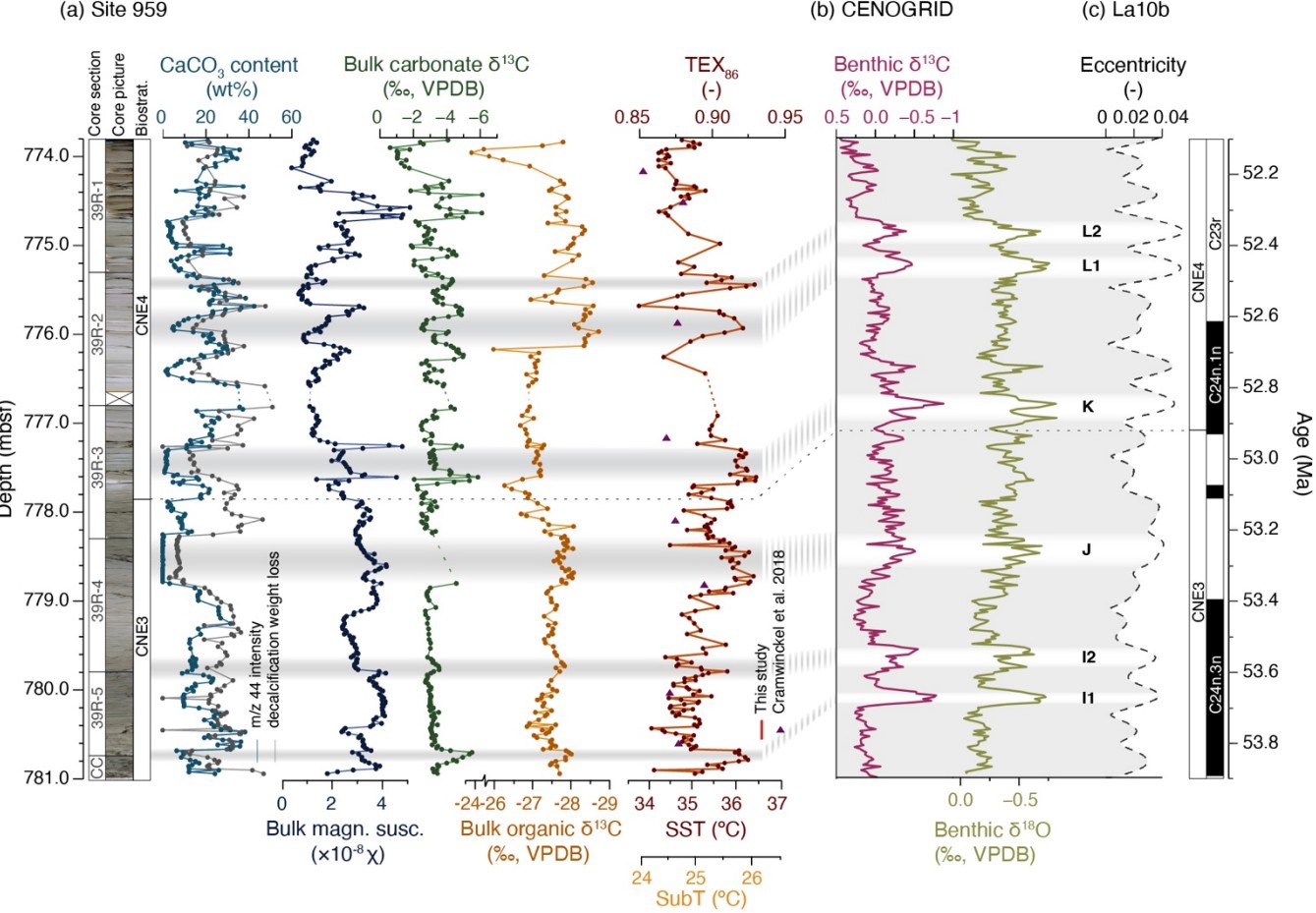

**Figure 4.** Downcore results of Site 959 Core 39R and correlation to benthic records and astronomical solution. (**a**) Results from Site 959 Core 39R against depth in mbsf. From left to right: core photographs (Mascle et al., 1996), calcareous nannofossil biostratigraphy (follows zonation by Agnini et al. (2014)), CaCO₃ content estimated from signal intensity during isotope ratio mass spectrometry analyses of bulk sediment (blue) and sediment weight loss after decalcification (grey), bulk MS, bulk carbonate δ¹³C, bulk organic δ¹³C and TEX₈₆. The pre-existing TEX₈₆ data of Cramwinckel et al. (2018) (n = 8) are shown in purple triangles. Horizontal bars indicate interpreted CIE horizons. (**b**) Deep ocean benthic foraminifer δ¹³C and δ¹⁸O from the CENOGRID compilation (Westerhold et al., 2020). (**c**) La10b astronomical eccentricity solution (Laskar et al., 2011), with the astronomically calibrated ages of CIEs I1, I2, J, K, L1 and L2 (Westerhold et al., 2017), and calcareous nannofossil zones of Agnini et al. (2014).

previously established early Eocene range of tropical-band SSTs based on $\delta^{18}O$, clumped isotopes and Mg/Ca paleothermometry of glassy foraminifera and $TEX_{86}$ (Evans et al., 2018; Cramwinckel et al., 2018; Gaskell et al., 2022). Our SST results are, however, generally higher compared to a previous $TEX_{86}$ study of the same Site (Fig. 4) (Cramwinckel et al., 2018). We attribute this offset to a sampling bias, as their analyses predominantly targeted darker sediment intervals for optimal GDGT preservation. Our data show that the darker intervals often coincide with slightly lower $TEX_{86}$ values (e.g., at ~778.2 and ~774.1; Fig. 4).

The $TEX_{86}$ record shows pronounced variability on orbital time scales, including warming across the various CIEs. As palynological analysis indicates an open ocean setting that was more oligotrophic than the remainder of the Eocene (Cramwinckel et al., 2018), with no discernable regional environmental variations (e.g., upwelling or terrestrial input) that would influence SSTs (Fig. S4), we assume that this reconstructed SST variability at Site 959 tracks the variability of the complete tropical band, in agreement with model simulations (Fig. 1b).

**3.3 Ice-free polar amplification of orbital-scale climate variability**

We first approximate short-term PA independently of direct stratigraphic correlation by a comparison of the SD of the Site 959 SST/SubT and open ocean BWT records, after detrending the dataset to remove Myr-scale trends. Over the interval of ~53.8–52 Ma, the SD of the detrended BWT record is 0.7 ºC versus 0.5 ºC of our tropical SSTs, suggesting that BWT variability was amplified with a factor of approximately 1.4 with respect to the equatorial SSTs (Fig. 5a). Naturally, this simple approach is sensitive to the different analytical errors of the temperature proxies. The analytical errors (~0.2 ºC for SST (see section 2.5) and ~0.4 ºC for $\delta^{18}O$-BWT (see section 2.7) are relatively large compared to the SDs of the complete records. However, the analytical errors are too small to explain the complete offset in variability of the two records. This implies that the obtained value is a crude approximation of PA. Performing the same exercise with SubT-calibrated $TEX_{86}$ data yields a SD of 0.4 ºC, and subsequently a higher amplification factor of ~1.8. Note that the higher PA estimate from SubT relative to SST does not imply that temperature variability was amplified more in the subsurface than in the surface ocean. Rather, calibrating equatorial $TEX_{86}$ values to a modern subsurface dataset yields a subdued $TEX_{86}$-temperature relation and therefore a lower reconstructed amplitude of past tropical temperature variability.

Next, we compare the magnitude of the recorded warming across negative CIEs, relative to the million-year background trend, between the tropics and deep sea (Fig. 5a–b). In comparison to deep ocean warming, events I1–L2 show a dampened response in the tropics. The dampened magnitude of tropical warming can be quantified as PA of climate change, with an amplification factor of 1.8±0.2 (SubT = 2.4±0.3). This amplification factor demonstrates that warming during these events was indeed amplified in the Southern Ocean surface waters, the presumed origin of early Eocene bottom waters, with respect to the tropical surface ocean. We compare these PA values to the PETM estimates, for which we use averaged previously published estimates on equatorial surface warming (ΔT = 2.9±0.5 ºC) (Frieling et al., 2017, 2019) and BWT reconstructions (ΔT = 5.4±0.5 ºC) (Dunkley Jones et al., 2013) (Fig. 4b). Importantly, the estimates for the PETM fall within the projected regression line of

events I1–L2, which indicates that, while the PETM was a more severe event in terms of climate change, PA was similar to the smaller events.

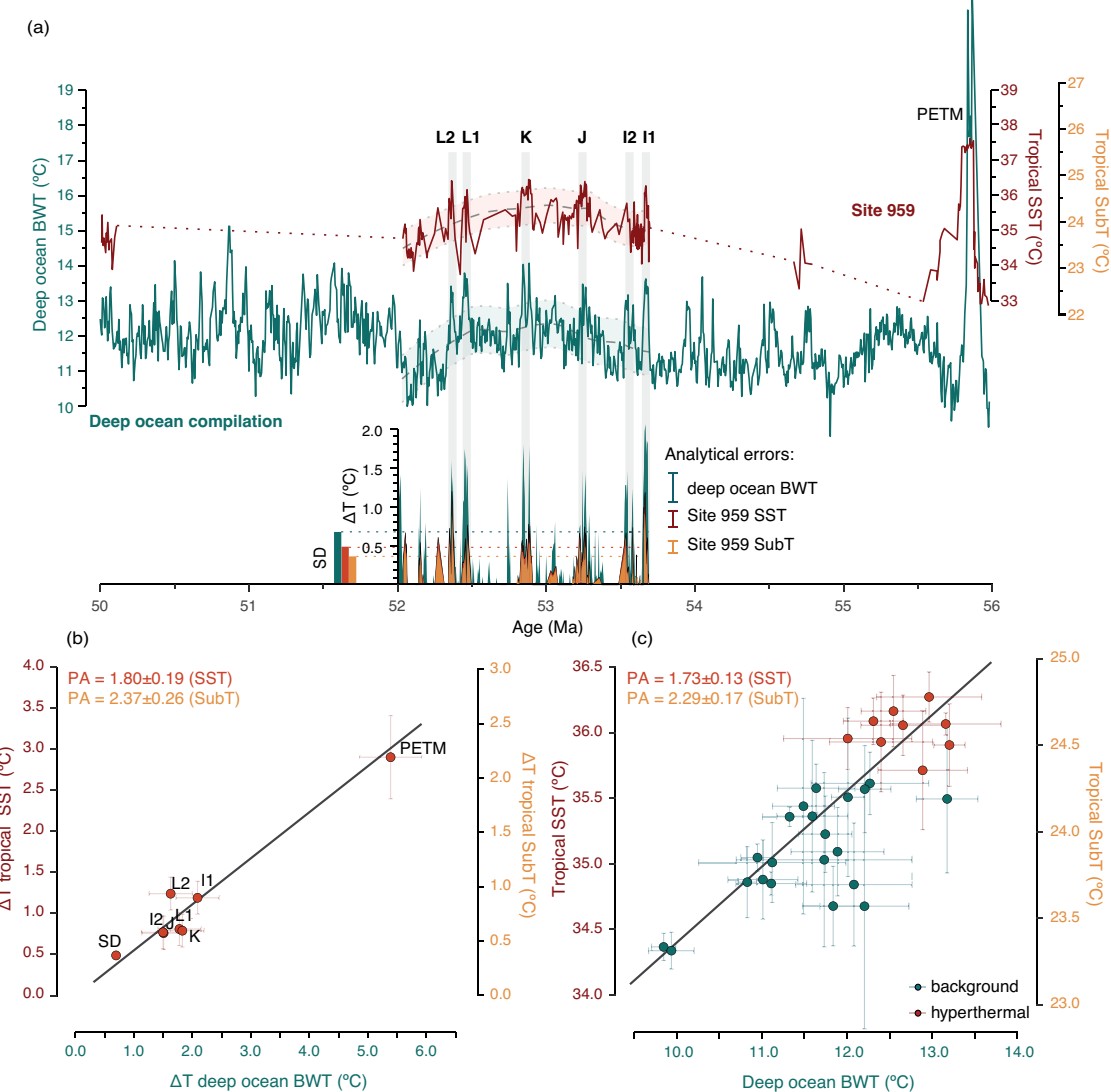

**Figure 5.** Comparison of temperature variability in the tropical (sub)surface and the deep ocean during the early Eocene. (**a**) Comparison of SST/SubT from Site 959 (red) and the CENOGRID benthic compilation (Westerhold et al., 2020) (green) (see section 2.7 for BWT calculations). The colored bands indicate the LOESS filter (window = 1 Myr) and 1 SD. Positive temperature deviations of deep ocean BWT (green), tropical SubT (yellow) and SST (red) relative to the 1-Myr trend are indicated below. CIEs are marked with their respective names above the record. Bars on the left indicate the values of SD per dataset, analytical errors are indicated on the right. (**b**) Magnitude of SST/SubT rise *versus* BWT rise across the multiple hyperthermals depicted in panel (**a**) (SST on left axis, SubT on right axis) and SD of the records as in bottom left panel (**a**). The errors (crosses) reflect analytical errors. The PETM data is from literature (Dunkley Jones et al., 2013; Frieling et al., 2017, 2019) with error crosses indicate standard error of combined estimates. The regression is only based on the warming of hyperthermals I1–L2. (**c**) Short-term, 20-kyr-binned Site 959 temperature dataset *versus* BWTs (Westerhold et al., 2020). The black lines in panels (**b**) and (**c**) represent the Deming regression lines between equatorial (sub)SSTs and BWTs, with respective PA factors on top left of the panels.

Our TEX$_{86}$ record varies in concert with the deep ocean BWT record (Westerhold et al., 2020) (Fig. 5) and shows that sub-million-year timescale climate variability was not limited to the globally recognized CIEs. Polar amplification of the complete, eccentricity-scale climate variability was evaluated by directly matching the SST record to the BWT record and dividing both records in 20-kyr bins (see section 2.7). For this detailed correlation, we included minor finetuning between related features in the benthic δ$^{18}$O record and TEX$_{86}$ records, so that warming events optimally line up (e.g., peak SSTs are compared to peak BWTs) (Fig. S7). Clearly, all other correlations are based on biostratigraphy and carbon isotope stratigraphy and therefore the positive relation between surface and deep ocean temperature variability is emergent. This observation warrants the additional correlations made here for the regression analysis on eccentricity scale to quantify polar amplification, but it assumes correspondence in the timing between surface and deep ocean temperature variability. A Deming regression using the binned data indicates a PA factor of ~1.7±0.1 (SubT = 2.3±0.2) between ~54 and ~52 Ma (Fig. 5c). Specifically, the PA derived from the short timespan included in the data bins demonstrates that climate change forced by 100-kyr eccentricity was amplified in the Southern Ocean during an ice-free climate state.

## 4. Discussion

### 4.1 I1, I2, J, K, L1 and L2 are hyperthermal events

The horizons of CIE events I1, I2, J, K, L1 and L2 all coincide with transient warmer intervals at Site 959, marked by a sea surface warming of approximately ~1 ºC (~0.7 ºC SubT) relative to background temperatures. Hence, the observed link between transient warming reconstructed in the tropics and deep ocean, together with the coeval deep ocean carbonate dissolution horizons (Leon-Rodriguez and Dickens, 2010; Westerhold et al., 2017), presents conclusive evidence that all these orbitally triggered carbon cycle perturbations were transient global warming events. As previously hypothesized (e.g., Cramer et al., 2003; Lourens et al., 2005), combined with the occurrence of CIEs and deep ocean carbonate dissolution, this evidence of global warming now proves that orbital forcing led to variability in atmospheric $p$CO$_2$ during the early Eocene ice-free world, due to carbon cycle feedbacks (e.g., Dickens, 2001; Setty et al., 2023).

### 4.2 Polar amplification during the ice-free EECO

Our reconstructed short-term (20-kyr) PA factor of 1.7–2.3 lies close to previous estimates on much longer, million-year timescales during a similar hothouse climate (Cramwinckel et al., 2018; Gaskell et al., 2022). To provide a more thorough comparison with 10$^6$-year timescale PA, we combine our high-resolution dataset with previously published late Paleocene–early Oligocene TEX$_{86}$-based SST data from Site 959 (Cramwinckel et al., 2018; Frieling et al., 2019) and compare to the deep ocean BWT dataset (Westerhold et al., 2020) in 1-Myr bins (Fig. 6a, yellow line). Remarkably, this 1-Myr-binned analysis results in a low to absent PA (~1.2±0.1) across the long-term Eocene cooling trend, that clearly differs from the 20-kyr-binned PA result during the early Eocene hyperthermals (Fig. 6a, red line). The small offset between our reconstructed SSTs and that

of Cramwinckel et al. (2018) (~1 ºC, Fig. 4) is insufficient to explain this difference in PA factors, especially as both datasets are combined for the long-term PA estimate in Fig. 6a. The discrepancy between long- and short-term PA could point to a timescale dependent PA, or alternatively to a (non-linear) GMST-dependent PA, as our short-term data is concentrated in the high temperature end. However, a change of local oceanographic conditions at Site 959 is a more likely explanation for the low PA in the long-term Site 959 PA estimate. Contrary to our high-resolution early Eocene interval (~54-52 Ma) (Fig. S4),

dinoflagellate cysts produced by heterotrophic taxa appear in the record around 49 Ma (Cramwinckel et al., 2018). In combination with the coeval increase in TOC, we interpret this as a progressive increase of surface water productivity at the Gulf of Guinea during the Eocene relative to the here-studied interval. At this location, increased productivity most likely relates to increased (seasonal) upwelling of cooler, nutrient-rich, sub-thermocline waters. Based on their records, Cramwinckel et al. (2018) assumed the presence of constant upwelling between ~58 and ~40 Ma at Site 959. However, specifically in the

here studied interval (~54–52 Ma), we find no evidence for upwelling in the dinocyst assemblages (Fig. S4) and this should have resulted in the higher temperatures we record in the early Eocene. This amplifies the apparent long-term cooling from the early to middle Eocene relative to the estimates of Cramwinckel et al. (2018), and reduces apparent PA. We therefore attribute the stronger cooling than Cramwinckel et al. (2018) recorded from the early to middle Eocene to reflect increased upwelling starting around 49 Ma. To reduce the effect of local cooling on reconstructed long-term PA, we compare our short-

term analysis to a long-term PA estimate that utilizes a more comprehensive tropical SST data compilation (Cramwinckel et al., 2018; Gaskell et al., 2022), including $TEX_{86}$, $\delta^{18}O$, Mg/Ca and $\Delta_{47}$ proxy records of 16 locations, to which we add our new dataset. A 1-Myr-binned comparison between this tropical SST compilation and deep ocean BWTs (Westerhold et al., 2020) results in a PA factor of ~1.6±0.1 (Fig. 6a, green line), which is within error of our short-term PA estimate. Coherence between the short- and long-term PA estimates implies that PA of the long-term $10^6$–$10^7$-year stochastic climate trends of the entire

largely ice-free Eocene is equal to PA on $10^4$-year climate variability of the early Eocene. Additionally, this correspondence implies that Eocene PA was not impacted by feedback mechanisms that act on $10^4$-year timescales or longer.

To test the capability of fully coupled climate models to accurately project ice-free PA, we compare our proxy-based estimates of PA to simulations from the DeepMIP ensemble that were run with multiple $CO_2$ forcings (Lunt et al., 2021). For optimal comparison, we use modelled low-latitude (30ºS–30ºN) annually averaged SSTs versus high-latitude Southern Ocean (>60º)

winter average SSTs, the presumed dominant source of Eocene bottom waters in these simulations (e.g., Huck et al. 2017; Hollis et al., 2012; Zhang et al., 2022) (Fig. 6a, grey line). The magnitude of combined-model-derived PA is ~1.2 and only small differences exist between individual model PA results, with highest PA value recorded by GFDL (PA = ~1.3) and smallest by COSMOS (PA = ~1.1). Utilization of tropical SSTs resulting from the $TEX_{86}^H$-SST calibration, which we consider to provide the most realistic SST calibration in our range of $TEX_{86}$ values (see section 2.6), shows the best correspondence of

short-term PA with PA from the model simulations, while using a SubT calibration results in larger PA factors. In contrast to the here used exponential calibrations, linear $TEX_{86}$ calibrations lead to much higher reconstructed tropical SST variability and consequently an absent or even negative PA factor.

Finally, although the climate models show that SST variability at the location of Site 959 should be an adequate tracer for SST variability in the complete tropical band (Fig. 1b), we stress the need for additional high resolution, early Eocene SST reconstructions. Specifically, new records from low latitudes would be of great value for validating and extending our record, and reconstructions from high latitudes will provide an optimal comparison for assessing PA.

The PA factor of the DeepMIP models essentially represents the non-ice-albedo PA of state-of-the art climate models. Therefore, the general agreement between our results and the magnitude of PA expressed by the DeepMIP output implies that (atmospheric) feedback processes to greenhouse gas forcing seem to be adequately represented in the DeepMIP model suite. However, our reconstruction points to somewhat larger PA in an ice-free world than that in the models, suggesting that the non-ice-related factors might be slightly underestimated in IPCC class current-generation fully coupled climate models. This is consistent with the commonly found result that many models underpredict Eocene PA and overestimate the MTGs. As cloud feedbacks present one of the largest uncertainties in climate models (Zelinka et al., 2022), and play an important role in ice-

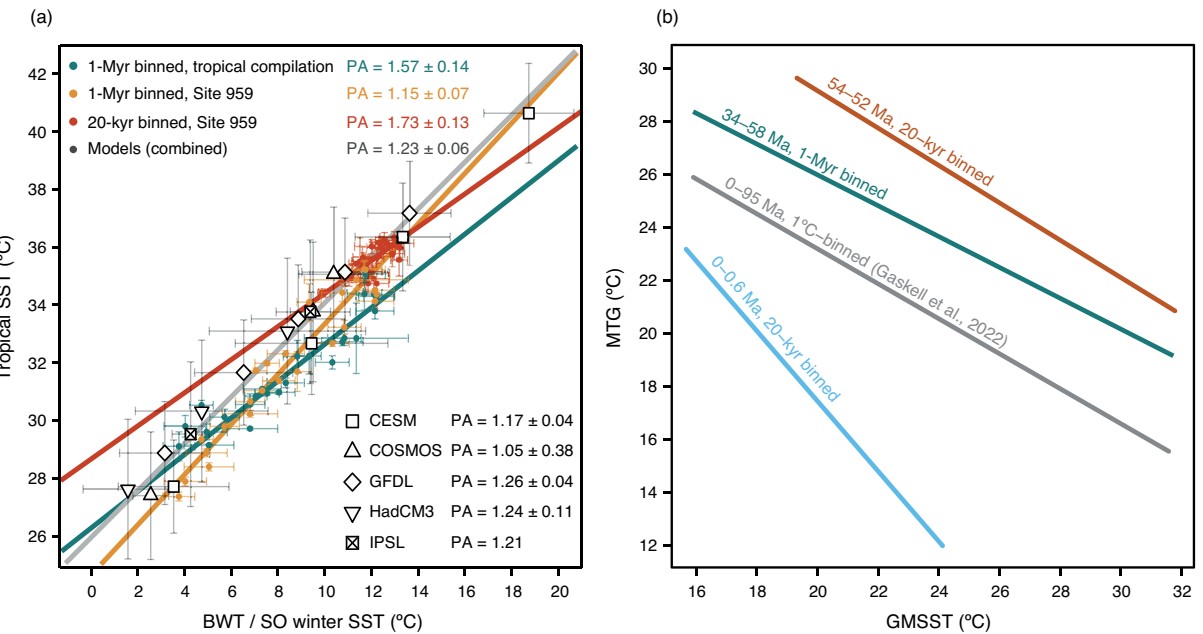

**Figure 6.** Polar amplification (**a**) and global mean sea surface temperature *versus* meridional temperature gradient (**b**) of various datasets and timespans. (**a**) Comparison of long-term (1-Myr bins), short-term (20-kyr bins) and modeled PA. In green, the 1-Myr binned tropical SST data compilation (Cramwinckel et al., 2018; Gaskell et al., 2022; This study) versus CENOGRID (Westerhold et al., 2020) BWTs, in yellow the 1-Myr binned complete Eocene Site 959 $TEX_{86}$-based SSTs (Cramwinckel et al., 2018; Frieling et al., 2019; This study) versus the CENOGRID BWTs and in red the 20-kyr binned Site 959 $TEX_{86}$-based SSTs versus the CENOGRID (as in Fig. 5c). Colored crosses represent uncertainty intervals based on binning (1 standard error). In grey, model output from a selection of different model simulations compiled from Lunt et al. (2021) (see section 2.7). Uncertainty crosses in the model datapoints represent one standard deviation across the selected latitudinal bands. All solid lines depict Deming regressions, based on the binned data and abovementioned uncertainty intervals. (**b**) GMSST versus ΔT Deming regression lines as calculated from 20-kyr binning between 54 and 52 Ma of Site 959 and CENOGRID (red), 1-Myr binning of tropical SST compilation and CENOGRID (green), temperature (1 °C) binning between high and low-latitude $\delta^{18}O$ data from 0 to 95 Ma (Gaskell et al., 2022) (grey) and 20-kyr binning the tropical (Herbert et al., 2010) and high-latitude (Lawrence et al., 2009; Martínez-Garcia et al., 2010; Ho et al., 2012) SST datasets from 0 to 0.6 Ma (blue).

free PA (Vavrus, 2004; Taylor et al., 2013a; England and Feldl, 2024), it is plausible that their effect is underestimated in the models. Indeed, several studies have shown that early Eocene high-latitude warmth can be better simulated by changing model cloud properties (Sagoo et al., 2013; Zhu et al., 2019). In addition, ocean heat transport, which is likely important for PA in ice-free climates (England and Feldl, 2024), is sensitive to changes in oceanic gateways or other factors such as orbital parameters and $CO_2$ concentrations (Huber and Nof, 2006), which might not be accurately represented in Eocene climate modeling (Zhang et al. 2022).

## 4.3 Eccentricity-forced global mean sea surface temperature

Our combined dataset of tropical SST and open ocean BWT approximates the range of warmest and coldest temperatures of the Eocene ocean, and can be utilized to estimate variations in both global mean sea surface temperature (GMSST) and MTG, by an area weighted average (Equation 1) (Caballero and Huber, 2013; Gaskell et al., 2022) and the difference between tropical SST and BWT, respectively. Intriguingly, the slope of the relationship between the MTG and GMSST is comparable between the 20-kyr binned dataset, the Myr-binned dataset and the $\delta^{18}$O-based data compilation from Gaskell et al. (2022) (Fig. 5b). Absolute GMSSTs are, however, offset by ~2 °C between our record and the Gaskell et al. (2022) dataset, likely due the different nature of the datasets, including general discrepancies in published $\delta^{18}$O records, inclusion of different sites, calibrations, and isotopic corrections (Fig. S3).

$$GMSST \ = \ 0.5 \times SST_{trop} \ + \ 0.366 \times \left( \frac{SST_{trop} - BWT}{2} \right) \ + \ 0.134 \times BWT \qquad\qquad (1)$$

Finally, we assess GMSST variability on eccentricity timescales (Fig. 6b, 7). Our analysis indicates that GMSST variability of recorded hyperthermals is ~1–1.5 ºC (using TEX$_{86}^{H}$) (Fig. 7a), which reveals that the modern GMSST warming of ~1 °C is already in the range of the early Eocene hyperthermal events. Importantly, we also record up to 0.7 °C variability on an approximate 100-kyr-eccentricity timescale in periods outside of hyperthermals (e.g., at ~53.0 and ~52.2 Ma in Fig 5a). We note, however, that some of our correlations between the tropical SST variations and the deep-ocean based on bio- and chemostratigraphy were finetuned using temperature proxies (Fig. S7), which might yield erroneous correlations of local temperature variations. Nevertheless, if observed GMSST variability indeed presents (100-kyr) eccentricity-paced global temperature variation, the Eocene ice-free climate responded strongly to small (~0.5 W/m$^2$) variations in global incoming solar radiation. This is strongly reminiscent of Pleistocene glacial-interglacial cycles (Herbert et al., 2010; Martínez-Garcia et al., 2010; Ho et al., 2012), but in the absence of strong amplifying ice and snow albedo feedbacks.

The eccentricity-forced GMSST variations during the early Eocene imply that there was a sensitive (organic) carbon cycle feedback mechanism at play, as suggested by previous work (Vervoort et al., 2021). It has previously been demonstrated that the benthic $\delta^{13}C$-$\delta^{18}O$ slope is equal during the hyperthermals and the background variations (Lauretano et al., 2015). This strongly suggests that the same carbon reservoirs and feedback mechanisms were responsible for the 405-kyr eccentricity-paced hyperthermals and the ~100-kyr eccentricity-paced climate variability. The proposed carbon capacitors responsible for hyperthermal variability include soils and peats (Kurtz et al., 2003), permafrost (DeConto et al., 2012) and methane hydrates (Dickens et al., 1997; Komar et al., 2013), all characterized by strong negative $\delta^{13}C$ signatures. Importantly, as the carbon release and warming is clearly of much greater magnitude during some of the larger, 405-kyr, eccentricity maxima, we can conclude that the characteristics of the responsible carbon cycle feedback must include that it can both scale with orbital

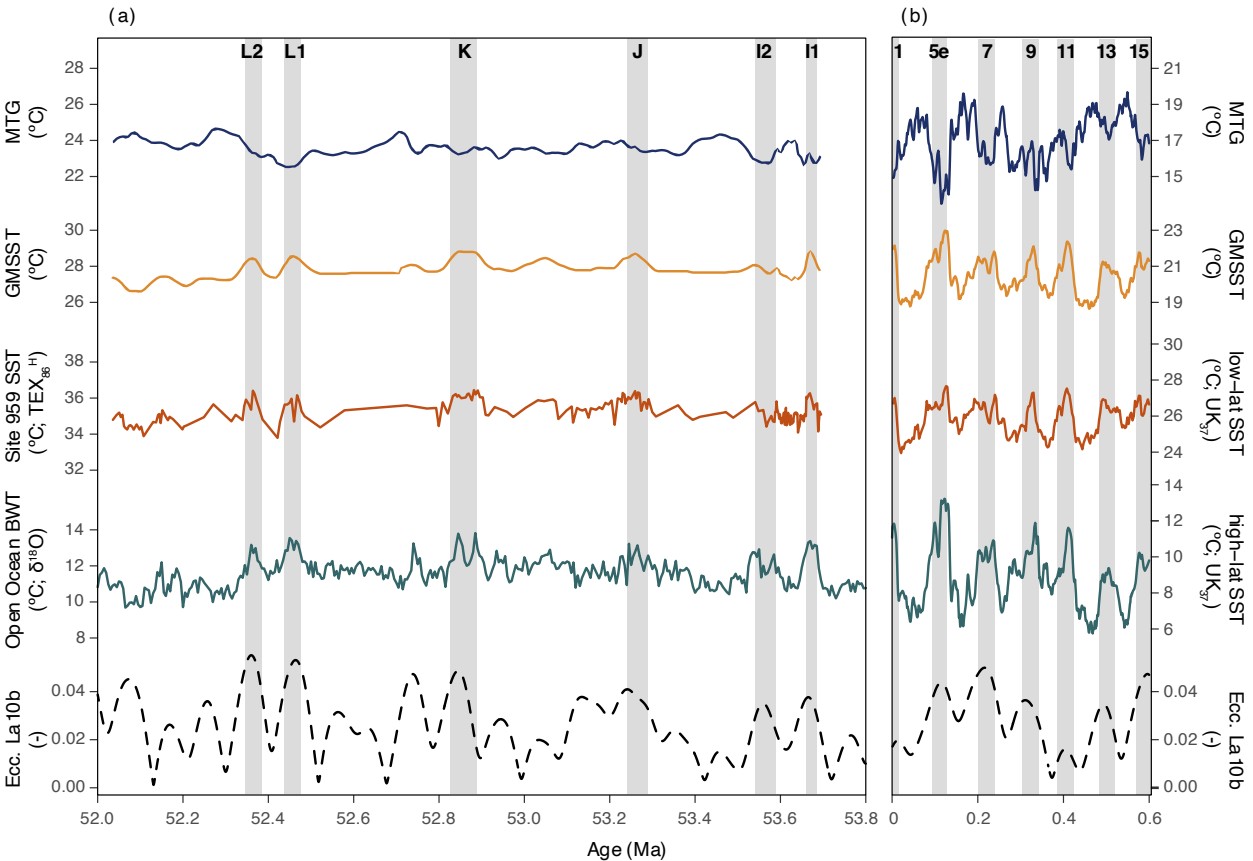

**Figure 7.** Global mean sea surface temperature, meridional temperature gradient and orbital eccentricity for the intervals 53.8–52 Ma (**a**) and 0.6-0 Ma (**b**). (**a**) Early Eocene data with tropical SST data from Site 959 (red) and open ocean BWTs from Westerhold et al. (2020) (green), estimated GMSST (yellow) and MTG (dark blue). (**b**) Late Pleistocene data with a $U^K_{37}$-based tropical SST compilation by Herbert et al. (2010) (red) and a $U^K_{37}$-based high-latitude SST dataset averaged from Lawrence et al., (2009); Martínez-Garcia et al. (2010); Ho et al. (2012) (green). Grey bars mark eccentricity related GMSST peaks, i.e. hyperthermals in the early Eocene and interglacials in the late Pleistocene with their respective hyperthermal name or MIS stage number on top. Eccentricity is from the La10b solution (Laskar et al., 2011).

forcing, and can accelerate after reaching a threshold or tipping point (Setty et al., 2023). Finally, although the exact mechanisms are uncertain, we note that the volume of $CO_2$ released via this (combination of) feedback mechanism(s) must be large. The logarithmic relation between $CO_2$ forcing and global mean temperature requires large volumes of $CO_2$ to force global temperature in the early Eocene high-$CO_2$ state (e.g., Lunt et al., 2021). A back-of-the-envelope calculation indicates

that, given an averaged modeled early Eocene Equilibrium Climate Sensitivity of 4.5 ºC per $pCO_2$ doubling (Lunt et al., 2021) and reconstructed background $CO_2$ concentrations of approximately ~1470 ppm (Anagnostou et al., 2020), recorded temperature variability would require a forcing of ~245 ppm $CO_2$ for the hyperthermals ($\Delta T = 1.0$ ºC) and ~118 ppm $CO_2$ for the other eccentricity-forced warmings ($\Delta T = 0.5$ ºC), broadly in line with previous estimates based on carbonate chemistry (Zeebe et al., 2017).

By means of comparison, we perform an additional GMSST estimate in terms of sampling and calculations for the most recent orbitally forced global temperature variations, which consist of the last few glacial-interglacial cycles (Fig. 7b). This dataset, composed of $U^K_{37}$ data from tropics (Herbert et al., 2010) and mid to high latitudes (Lawrence et al., 2009; Martínez-Garcia et al., 2010; Ho et al., 2012) shows GMSST variations of ~4 ºC, similar to a recent estimate (Annan et al., 2022), as well as a stronger PA than during the early Eocene (Fig. 5b). A general assumption for global climate over glacial-interglacial cycles is

that they are largely facilitated by ice-related positive climate feedback mechanisms (Osman et al., 2021). Our data from the ice-free early Eocene, however, imply that eccentricity forcing, through positive climate feedbacks related to the carbon cycle, was capable of producing GMSST variation roughly one third the amplitude of glacial-interglacial variability. The different background conditions of the early Eocene compared to the Pleistocene, which, apart from significant differences in ice-related surface albedo, includes higher background temperatures, higher sea level and a different paleogeography, could have caused

certain carbon cycle feedback mechanisms to act differently than in the Pleistocene. For example, it can be assumed that the warm Eocene climate would have largely restricted the area of permafrost in the Antarctic interior that experienced relatively warm and wet summers (Baatsen et al., 2024). Intriguingly, the carbon storage in permafrost may have been replaced by extensive peat deposits with, presumably, a similar carbon cycle impact. Moreover, the higher seafloor temperatures would greatly reduce the potential volume of methane hydrates (Dickens, 2001b), although a long-term warming since the Paleocene

might have put the methane hydrates closer to a critical threshold (Zachos et al., 2001). Nevertheless, the strong eccentricity imprint raises the question if similar poorly constrained carbon cycle feedbacks, that do not involve ice, snow and frost-related processes, were only inherent to past greenhouse climates, or if they also played a role in Pleistocene glacial-interglacial climate variability.

**5. Conclusions**

Our new high-resolution SST dataset from the early Eocene tropics confirms that several deep ocean foraminifer $\delta^{13}C$ and $\delta^{18}O$ isotope events were associated with tropical warming and therefore represent transient global warming events (hyperthermals). The record also shows that SST variability in the tropics was smaller than at high latitudes on timescales of orbital eccentricity.

The resulting estimate of PA is 1.7–2.3, within error of previous proxy data-based estimates over longer timespans. Because a major surface albedo feedback contribution from ice can be largely ruled out in our new short-term dataset, we conclude that early Eocene PA is dominated by non-ice feedback mechanisms that act on $10^4$-year timescales or shorter. Earth system model simulations generally capture these climate feedbacks relatively well but may underestimate the strength of non-ice albedo feedbacks and thereby PA in ice-free climate states. Combined with the deep ocean BWT records, our data suggests eccentricity-forced GMSST changes of up to ~1 °C — even outside the hyperthermals — in the ice-free early Eocene. Such variability in tropical SST and GMSST necessitates very strong carbon cycle feedbacks to orbital forcing during that time that may well have been active throughout geological time.

## Data availability

All data used in this study is published on Zenodo (https://doi.org/10.5281/zenodo.8309643).

## Author contributions

CDF and AS designed the research. CDF, DG, LV, MdG, PdR and TA carried out bulk sediment analyses. CA performed calcareous nannofossil paleontology. CDF, FP, TA and AR analyzed lipid biomarkers. CDF, DG, MdG, PdR and TA performed palynology. CDF and XL carried out model comparisons. CDF carried out the data analysis. CDF and AS wrote the paper with contributions from MH, XL, PKB, FP and JF.

## Acknowledgements

This project used samples and data provided by the International Ocean Discovery Program (IODP) and predecessors. We thank A. van den Dikkenberg, G. Dammers, N. Welters, A. van Dijk, D. Eefting and M. Krasnoperov (all Utrecht University) for technical and analytical assistance. We thank H. Kuhlmann and A. Wülbers for support during our (many) unusual sampling campaigns at Bremen Core Repository (BCR). Furthermore, we thank U. Röhl and T. Westerhold (MARUM, Bremen) for discussions regarding Site 959 stratigraphy. This project is funded by European Research Council Consolidator Grant 771497 awarded to AS under the Horizon 2020 program and benefitted from intellectual contributions by members of the Netherlands Earth System Science Centre, funded by Gravitation Grant 024.002.001 from the Dutch Ministry of Education, Culture and Science. MH and XL were funded by an NSF grant OPP-1842059 awarded to MH. Financial support to CA was provided through the PRIN (Prot. 2022T4XEBP) and the extended partnership RETURN, financed by the National Recovery and Resilience Plan (NRRP), Mission 4, Component 2, Investment 1.3–D.D. 1243 of 2/8/2022, PE0000005. We greatly appreciate detailed comments from two anonymous reviewers that helped improve this work and editor Zhengtang Guo for handling.

## Competing interests

One of the co-authors is a member of the editorial board of Climate of the Past.

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
