# Peer review of "Polar amplification of orbital-scale climate variability in the early Eocene greenhouse world"

_Climate of the Past, 2023_

## Referee Comment (RC1)

**Overarching comments and appreciation of the study (manuscript text)**

In their study, Fokkema et al. study polar amplification (PA) of orbital-scale climate variability at a time of the early Cenozoic, when ice sheets were much less wide-spread than they are now, even absent. In comparison to studying Pleistocene climate variability and amplitude of PA, this setting allows separation of the impact of ice-related feedbacks on PA from non-ice-related mechanisms. The authors stress that a big step in their work is the construction of a multi-millenial data set of variability of tropical sea surface temperature. The authors describe the derivation of quantitative inference from cored sediment material at Site 959, discuss and define a calibration to the temperature derivation, and describe an age model that is refined based on previous work. Reliability of climate signals is discussed in detail, as a result of which delta18O is excluded from further analyses.

In their supplement, the authors present a concise overview on the calibration of sea water temperature to the geologic archive TEX[86] that they use, arguing that the use of an exponential calibration is more suited to represent tropical temperatures of warm climates like the Paleogene. I find this argumentation quite important and suggest to move it to the manuscript text.

The work by the authors, that led to this study, is greatly appreciated. For example, this work enables testing the ability of climate models to reproduce PA as recorded in the geologic archive - a valuable opportunity as PA is one of the relevant climate system metrics for understanding and projecting future climate at much warmer than modern high latitudes. This work may hence extend our model validation from the very short modern observational period towards past (future-analogue) climates. The utility of the presented data towards exactly this purpose is demonstrated by the authors at the example of PA in an ice-free EECO climate. Based on a comparison of their sea surface temperature reconstruction with DeepMIP model simulations presented by Lunt et al. (2021), the authors infer that models agree well with inferences from the geologic record, while noting that PA-causing processes and mechanisms, aside from those related to ice, may be underestimated in the models.

The authors present various insights that are very valuable. Their work supports linkages between climate variability of an early Cenozoic hothouse world with dynamics on glacial-interglacial cycles, posing the question to which extent carbon-cycle feedbacks played a role in Pleistocene Milancovitch cycles. Furthermore, the work illustrates the magnitude of current climate change in the context of Cenozoic climate history. I find the statement „modern GMSST warming of ~1 °C is already in the range of the early Eocene hyperthermal events" particularly remarkable.

In my opinion the manuscript is very well, and carefully, written. I have located several minor issues and provide some comments at locations where I, as an interested reader, would like to have a bit more insight regarding specific aspects of the work. One addition to the discussion / conclusion outlook could be to propose testing whether the findings of this study will be reproduced in analyses from other cores and from other core locations at a similar latitude. I have no reason to doubt the assumptions made by the authors, regarding both spatial representativeness of the reconstructed signal of sea surface temperature variability and depth of the water column to which a reconstructed temperature signal is attributed; and, as the authors note, their assumptions are supported by auxiliary evidence, as for example by climate modelling. Nevertheless, further testing of these assumptions based on material from other cores is, at least in my opinion, worthwhile. Spending one or two related sentences in the discussion, and/or in an outlook section, would in my opinion provide a meaningful conclusion to the manuscript.

I note that I am not a climate scientist conducting analyses of the sedimentary records similar to those analyses described by the authors. Hence, my knowledge in details of sample preparation procedures and analysis methodology is not very deep. If the editors have any doubts regarding the validity of the analysis methodology, then I kindly ask to refer to an expert in that field for a second opinion.

In summary, I support publication of the study in Climate of the Past subsequent to addressing or rebuting comments.

**Specific comments (main text)**

Terminology regarding geologic timescales: I noted that in their supplement the authors refer several times to Paleogene or early Paleogene, while the term barely appears in the main text. When the authors speak of the Early Paleogene, do they refer to the (early) Eocene that is often referred to in the main text? I do not think that the link is always clear - in particular, to my understanding, the early Paleogene would likely rather refer to the Paleocene than to the Eocene? I may be wrong here, but clarifying the text where necessary may be helpful for readers.

Line 24: add comma after „orbital"

Line 44: add comma after „feedbacks to PA"

Lines 40-49: re origin of PA in climate models: I suggest to also refer to / comment on the latitude-dependency of the fraction of outgoing radiation (Pithan and Mauritsen, 2014) via temperature feedbacks, as these have been found to dominate the mechanism for PA in CMIP5 models.

Line 102: Meaning of the text „no ice and continental configuration" remains unclear. Do the authors aim at the degree of detail of paleogeography considered in the Eocene simulations? If so, should this text rather read „adaptation of model geography to reconstructed continental configuration and absence of major ice sheets" or similar?

Line 154: add „for" after 0.07‰

Line 156: remove space between 13 and %

Line 158: were -> was

Line 164: remove spaces between values and ‰ and % signs, respectively

Line 219: I am not sure whether the term „neutralize" is correct here. Tap water is not neutralizing afaik, it is rather diluting - does'nt one need to add a base in order to neutralize an acid?

Line 252: change to „a climate signal" or „as a climate signal"

Line 261: fix the format of the DOI? (remove the space and/or make the doi a hyperlink?)

Line 264: plots of CENOGRID are at least to me confusing due to the same / very similar color being chosen for both benthic d13C and benthic d18O. Based on the alignment of data and y-axis I can guess which branch represents which isotopic ratio, but the color coding is not helpful here. The „bracket"-like signal on the right, near 56 Myr, is unclear to me - please explain if relevant.

Line 272: Maybe provide the modern temperatures in the region as a reference for the 35.2°C of Eocene SST towards providing a rough estimate of climatic difference wrt. to today?

Page 10, Fig. 2: Clarify the meaning of „E-08" of the susceptibility record - shall this be $10^{-8}$? Refer to my comment in Fig. 1 regarding colors of CENOGRID records. I assume the reference (c) in the figure caption should be moved from the end of the sentence to before „Calcareous nannofossil zones"?

Line 282: „The record shows" or „The records show"

Line 286: Refer to my suggestion to move much of the information from the supplement into the manuscript text. Yet, even if this is not done, my feeling is that a bit more information regarding the model simulations should be given here than just a reference to the supplement.

Line 288: „of both records": Maybe once more explicitly state which records you refer to, for clarity. Same for „the dataset".

Line 289: Should SD be in singular here? If plural, maybe there is a problem with the formulation.

Line 292: Indeed, the calibration uncertainty / analytical errors are small regarding the absolute values, but they are large in comparison to the reconstructed SD?

Line 325: Fig. 3 suggests that especially CIE J and K coincide with a state of relative warmth rather than a state or warming, that may even include a subsequent cooling, but not necessarily only a phase of transient warming - am I right with this observation?

Line 365: Do the authors refer to the whole tropical band or only to the Northern Hemisphere part? Furthermore, regarding the statement „the dominant source of Eocene bottom waters in these simulations" - is this an inference that authors make based on their own analyses, or do the authors refer to results by Lunt et al. (2021), or maybe even to results from authors who contributed model simulations towards the model intercomparison by Lunt et al. (2021)? If the result is not derived by the authors themselves, then I assume citing the relevant publication(s) in this specific context makes sense.

Figure 5, caption: fix bracket of (Herbert et al., 2010)

Lines 416-418: Simplify reading by adding some commas: „[...] carbon cycle feedbacks, that do not involve ice, snow and frost-related processes, were only inherent to past greenhouse climate, [...]"

Lines 424-426: please check the sentence „[...] we conclude that early Eocene PA is not impacted by non-ice feedback mechanisms that act on $10^4$-year timescales or longer."

**Specific comments (Supplementary material):**

In my opinion the supplementary material present information that is key to a full appreciation of the work presented by the authors. In my opinion the text on pages 2-6 is actually very relevant for a deeper understanding of the work. In particular section 1.1, but also the other sections, would in my humble opinion fit well into the manuscript text. If there is no good reason to put this text into a supplement, I would suggest to move it to the manuscript. I think the description of employed model output (lines 154-159) should really be presented in a data section to make the link between proxy data analysis and climate modeling more clear in the manuscript. In this context please highlight that the simulations by Lunt et al. (2021) represent climate states of the early Eocene climate optimum (EECO, ~ 50 million years ago). Is there any need to „extrapolate" results derived from these simulations to different periods described in the manuscript, in particularly an early Paleogene climate (I am not quite sure about the definition of early Paleogene)?

Line 22-24: I think the text is potentially ambiguous, should it read as follows?: „Following the original linear TEX86-sea surface temperature (SST) calibration (Schouten et al., 2002), subsequently proposed calibrations include linear (O'Brien et al., 2017) models, including a spatially varying Bayesian approach ('BAYSPAR') (Tierney and Tingley, 2014), and as well reciprocal (Liu et al., 2009) and exponential (Kim et al., 2010) models.“

Line 77-82: Assuming for a moment that the thickness of the early Paleogene mixed layer might have been generally different from today, for example as a result of different intensity of stirring of the upper ocean layers due to invigorated atmosphere dynamics: how would a different water column structure of the early Paleogene tropical Pacific, in particular a different thickness of the mixed layer, impact on the calibration of the target depth, and potentially on results and inferences drawn by the authors in this work? There seems to be evidence that a different thickness of the mixed layer depth cannot be excluded (Quillevere and Norris, 2003; Barnet et al., 2020). Would the impact on peak integrated GDGT source depth be relevant, or are there indications that the effect would be negligible?

Line 90: fix brackets of (Ho and Laepple, 2016)

Line 114: fix brackets of (Kim and O'Neil, 1997)

Line 155: fix brackets of (Lunt et al., 2021):

Line 175: fix „Dashed red dashed line“

Line 188: define CIE

Line 196: Is there a specific reason for bold-typesetting the two publications? Fix brackets of (Miller et al., 2020).

Line 202: captialize after (c) and fix formatting of d13Corg

Line 211: Fix formatting of (Cramwinckel et al., 2018; Frieling et al., 2019; This Study) (this text is dashed underlined, it is not clear whether this is on purpose)

Line 208: Fig. S7 (a-d): Do I correctly interpret that all data points from this site (grey) and from this study (black triangles), are located exactly on the calibration line in subfigures c and d, without any kind of deviation that is apparent to the eye? My apology if I overlooked something obvious, but is there an explanation for this fact? I suggest to give the details of the calibration model, e.g. r-value and fit equation as done for Fig. S1.

Line 225, Fig. S9: Black and grey dashed lines in a, b, and c look the same to me (insufficient color contrast). Are different colors needed here? If not, just use one color.

**References:**

Quillëvërë, F., and Norris, R. D.: "Ecological development of acarininids (planktonic foraminifera) and hydrographic evolution of Paleocene surface waters", in Causes and consequences of globally warm climates in the early Paleogene, Scott L. Wing, Philip D. Gingerich, Birger Schmitz, Ellen Thomas, 2003.

Barnet, J.S.K., Harper, D. T., LeVay, L. J., Edgar, K. M., Henehan, M. J., Babila, T. L., Ullmann, C. V., Leng, M. J., Kroon, D., Zachos, J. C., and Littler, K.: Coupled evolution of temperature and carbonate chemistry during the Paleocene–Eocene; new trace element records from the low latitude Indian Ocean, Earth and Planetary Science Letters, 545, 116414, https://doi.org/10.1016/j.epsl.2020.116414, 2020.

Lunt, D. J., Bragg, F., Chan, W.-L., Hutchinson, D. K., Ladant, J.-B., Morozova, P., Niezgodzki, I., Steinig, S., Zhang, Z., Zhu, J., Abe-Ouchi, A., Anagnostou, E., de Boer, A. M., Coxall, H. K., Donnadieu, Y., Foster, G., Inglis, G. N., Knorr, G., Langebroek, P. M., Lear, C. H., Lohmann, G., Poulsen, C. J., Sepulchre, P., Tierney, J. E., Valdes, P. J., Volodin, E. M., Dunkley Jones, T., Hollis, C. J., Huber, M., and Otto-Bliesner, B. L.: DeepMIP: model intercomparison of early Eocene climatic optimum (EECO) large-scale climate features and comparison with proxy data, Clim. Past, 17, 203–227, https://doi.org/10.5194/cp-17-203-2021, 2021.

Pithan, F., Mauritsen, T. Arctic amplification dominated by temperature feedbacks in contemporary climate models. Nature Geosci 7, 181–184 (2014). https://doi.org/10.1038/ngeo2071

---

## Author Comment (AC1)

Dear Prof. Guo,

We greatly appreciate the highly detailed and very positive evaluation of our work by Reviewer #1 and the nice words they convey regarding our intention and outcome of the study.

Below we reply to each point they raise and indicate how we intend to revise the manuscript.

Sincerely, also on behalf of all co-authors,

Chris Fokkema

**Overarching comments and appreciation of the study (manuscript text)**

In their study, Fokkema et al. study polar amplification (PA) of orbital-scale climate variability at a time of the early Cenozoic, when ice sheets were much less wide-spread than they are now, even absent. In comparison to studying Pleistocene climate variability and amplitude of PA, this setting allows separation of the impact of ice-related feedbacks on PA from non-ice-related mechanisms. The authors stress that a big step in their work is the construction of a multi-millenial data set of variability of tropical sea surface temperature. The authors describe the derivation of quantitative inference from cored sediment material at Site 959, discuss and define a calibration to the temperature derivation, and describe an age model that is refined based on previous work. Reliability of climate signals is discussed in detail, as a result of which delta18O is excluded from further analyses.

In their supplement, the authors present a concise overview on the calibration of sea water temperature to the geologic archive TEX$^{86}$ that they use, arguing that the use of an exponential calibration is more suited to represent tropical temperatures of warm climates like the Paleogene. I find this argumentation quite important and suggest to move it to the manuscript text.

**Author response:** We thank the reviewer for this suggestion and will transfer the Supplementary Text to the main text.

The work by the authors, that led to this study, is greatly appreciated. For example, this work enables testing the ability of climate models to reproduce PA as recorded in the geologic archive - a valuable opportunity as PA is one of the relevant climate system metrics for understanding and projecting future climate at much warmer than modern high latitudes. This work may hence extend our model validation from the very short modern observational period towards past (future- analogue) climates. The utility of the presented data towards exactly this purpose is demonstrated by the authors at the example of PA in an ice-free EECO climate. Based on a comparison of their sea surface temperature reconstruction with DeepMIP model simulations presented by Lunt et al. (2021), the authors infer that models agree well with inferences from the geologic record, while noting that PA-causing processes and mechanisms, aside from those related to ice, may be underestimated in the models.

The authors present various insights that are very valuable. Their work supports linkages between climate variability of an early Cenozoic hothouse world with dynamics on glacial-interglacial cycles, posing the question to which extent carbon-cycle feedbacks played a role in Pleistocene Milancovitch cycles. Furthermore, the work illustrates the magnitude of current climate change in the context of Cenozoic climate history. I find the statement „modern GMSST warming of ~1 °C is already in the range of the early Eocene hyperthermal events" particularly remarkable.

In my opinion the manuscript is very well, and carefully, written. I have located several minor issues and provide some comments at locations where I, as an interested reader, would like to have a bit more insight regarding specific aspects of the work. One addition to the discussion / conclusion outlook could be to propose testing whether the findings of this study will be reproduced in analyses from other cores and from other core locations at a similar latitude. I have no reason to doubt the assumptions made by the authors, regarding both spatial representativeness of the reconstructed signal of sea surface temperature variability and depth of the water column to which a reconstructed temperature signal is attributed; and, as the authors note, their assumptions are supported by auxiliary evidence, as for example by climate modelling. Nevertheless, further testing of these assumptions based on material from other cores is, at least in my opinion, worthwhile. Spending one or two related sentences in the discussion, and/or in an outlook section, would in my opinion provide a meaningful conclusion to the manuscript.

**Author response:** We thank the reviewer for this recommendation. We will include such a statement in the discussion, where we will highlight the need for additional high resolution, early Eocene SST reconstructions. Low latitude records for validating and extending our work, and high latitude records to provide an optimal comparison for assessing PA.

I note that I am not a climate scientist conducting analyses of the sedimentary records similar to those analyses described by the authors. Hence, my knowledge in details of sample preparation procedures and analysis methodology is not very deep. If the editors have any doubts regarding the validity of the analysis methodology, then I kindly ask to refer to an expert in that field for a second opinion.

In summary, I support publication of the study in Climate of the Past subsequent to addressing or rebuting comments.

**Author response:** We thank the reviewer for their appreciation of our work.

**Specific comments (main text)**

Terminology regarding geologic timescales: I noted that in their supplement the authors refer several times to Paleogene or early Paleogene, while the term barely appears in the main text. When the authors speak of the Early Paleogene, do they refer to the (early) Eocene that is often referred to in the main text? I do not think that the link is always clear - in particular, to my understanding, the early Paleogene would likely rather refer to the Paleocene than to the Eocene? I may be wrong here, but clarifying the text where necessary may be helpful for readers.

**Author response:** We realize this might be confusing for members of our community that are not familiar to the details of stratigraphy and stratigraphical nomenclature. For clarity, "Paleogene" will be changed to "Eocene" to remain consistent in the terminolgy.

Line 24: add comma after „orbital"

Line 44: add comma after „feedbacks to PA"

**Author response:** Commas will be added.

Lines 40-49: re origin of PA in climate models: I suggest to also refer to / comment on the latitude-dependency of the fraction of outgoing radiation (Pithan and Mauritsen, 2014) via temperature feedbacks, as these have been found to dominate the mechanism for PA in CMIP5 models.

**Author response:** This will be done.

Line 102: Meaning of the text „no ice and continental configuration" remains unclear. Do the authors aim at the degree of detail of paleogeography considered in the Eocene simulations? If so, should this text rather read „adaptation of model geography to reconstructed continental configuration and absence of major ice sheets" or similar?

**Author response:** This will be clarified to "(i.e., early Eocene paleogeography without ice sheets)"

Line 154: add „for" after 0.07‰

Line 156: remove space between 13 and %

Line 158: were -> was

Line 164: remove spaces between values and ‰ and % signs, respectively

Line 219: I am not sure whether the term „neutralize" is correct here. Tap water is not neutralizing afaik, it is rather diluting - does'nt one need to add a base in order to neutralize an acid?

**Author response:** Above textual errors will be fixed, and "neutralized" (line 219) will be changed to "diluted"

Line 252: change to „a climate signal" or „as a climate signal"

**Author response:** This will be done

Line 261: fix the format of the DOI? (remove the space and/or make the doi a hyperlink?)

**Author response:** this will be fixed.

Line 264: plots of CENOGRID are at least to me confusing due to the same / very similar color being chosen for both benthic d13C and benthic d18O. Based on the alignment of data and y-axis I can guess which branch represents which isotopic ratio, but the color coding is not helpful here. The „bracket"- like signal on the right, near 56 Myr, is unclear to me - please explain if relevant.

**Author response:** We will change the colors and flipping/ resizing axes to solve these problems.

Line 272: Maybe provide the modern temperatures in the region as a reference for the 35.2°C of Eocene SST towards providing a rough estimate of climatic difference wrt. to today?

**Author response:** We will add the modern regional SSTs (annual average of 27.7 ºC Locarnini et al. 2018 (*WOAA*).

Page 10, Fig. 2: Clarify the meaning of „E-08" of the susceptibility record - shall this be $10^{-8}$? Refer to my comment in Fig. 1 regarding colors of CENOGRID records. I assume the reference (c) in the figure caption should be moved from the end of the sentence to before „Calcareous nannofossil zones"?

**Author response:** "E-08" will be changed to "$\times 10^{-8}$". "(c)" will be moved to the front of the respective sentence.

Line 282: „The record shows" or „The records show"

**Author response:** "The records shows" will be adjusted to "The $TEX_{86}$ record shows"

Line 286: Refer to my suggestion to move much of the information from the supplement into the manuscript text. Yet, even if this is not done, my feeling is that a bit more information regarding the model simulations should be given here than just a reference to the supplement.

**Author response:** We agree, and will move the relevant parts of the supplement to the main text, include the information about the model simulations.

Line 288: „of both records": Maybe once more explicitly state which records you refer to, for clarity. Same for „the dataset".

**Author response:** This will be done.

Line 289: Should SD be in singular here? If plural, maybe there is a problem with the formulation.

**Author response:** "SDs" will be changed to "SD"

Line 292: Indeed, the calibration uncertainty / analytical errors are small regarding the absolute values, but they are large in comparison to the reconstructed SD?

**Author response:** We describe here that the analytical uncertainty on the recorded variability is smaller than the signal (SD), and the calibration error to absolute water temperature values is much greater. We will reconsider wording to optimize clarity in such a way that the shortcomings of simply comparing SDs, in particular related to the analytical errors, will be better emphasized. Note, however, that this method of comparing SDs is only included as a simple, first-order, approach of comparing the magnitude of variability of the two records,without any further stratigraphic correlations.

Line 325: Fig. 3 suggests that especially CIE J and K coincide with a state of relative warmth rather than a state or warming, that may even include a subsequent cooling, but not necessarily only a phase of transient warming - am I right with this observation?

**Author response:** For the hyperthermals we use the wording "transient sea surface warming" to describe the complete event, i.e. the period 'above background temperatures' that includes the initial and peak and generally also a return to background temperatures. We will clarify this by describing the temperature effects during the events here as "warmer intervals" .

Line 365: Do the authors refer to the whole tropical band or only to the Northern Hemisphere part?

**Author response:** We here refer to the complete low-latitude band, and this will be clarified by changing "<30º" to "30º S – 30º N".

Furthermore, regarding the statement „the dominant source of Eocene bottom waters in these simulations" - is this an inference that authors make based on their own analyses, or do the authors refer to results by Lunt et al. (2021), or maybe even to results from authors who contributed model simulations towards the model intercomparison by Lunt et al. (2021)? If the result is not derived by the authors themselves, then I assume citing the relevant publication(s) in this specific context makes sense.

**Author response:** The statement is based on data and model-based inferences as described in the introduction: published by (amongst others) Cramwinckel et al. 2018 (*Nature*), Gaskell et al. 2022 (*PNAS*), Hollis et al. 2012 (*EPSL*), Zhang et al. 2022 (*P&P*). For clarity, we will add references to this statement.

Figure 5, caption: fix bracket of (Herbert et al., 2010)

**Author response:** The bracket will be fixed.

Lines 416-418: Simplify reading by adding some commas: „[...] carbon cycle feedbacks, that do not involve ice, snow and frost-related processes, were only inherent to past greenhouse climate, [...]"

**Author response:** Commas will be added.

Lines 424-426: please check the sentence „[...] we conclude that early Eocene PA is not impacted by non-ice feedback mechanisms that act on $10^4$-year timescales or longer."

**Author response:** Will be changed to: " Eocene PA is dominated by non-ice feedback mechanisms that act on $10^4$-year timescales or shorter

**Specific comments (Supplementary material):**

In my opinion the supplementary material present information that is key to a full appreciation of the work presented by the authors. In my opinion the text on pages 2-6 is actually very relevant for a deeper understanding of the work. In particular section 1.1, but also the other sections, would in my humble opinion fit well into the manuscript text. If there is no good reason to put this text into a supplement, I would suggest to move it to the manuscript. I think the description of employed model output (lines 154-159) should really be presented in a data section to make the link between proxy data analysis and climate modeling more clear in the manuscript.

**Author response:** We agree with this point by the reviewer. We will move all the supplementary text to the main text, including the section on $TEX_{86}$ and model outputs. Additionally, we will change "early Paleogene" to "(early) Eocene", to be consistent with the main text, as this encompasses all the intended stratigraphic range.

In this context please highlight that the simulations by Lunt et al. (2021) represent climate states of the early Eocene climate optimum (EECO, ~ 50 million years ago). Is there any need to „extrapolate" results derived from these simulations to different periods described in the manuscript, in particularly an early Paleogene climate (I am not quite sure about the definition of early Paleogene)?

**Author response:** Our dataset covers the onset of the EECO (*ca.* 53 – 49 Ma, e.g. Westerhold et al. 2018, *P&P*), and match the target of DeepMIP simulations that we compare to (i.e., the boundary conditions regarding ice sheets and continental configuration).

Line 22-24: I think the text is potentially ambiguous, should it read as follows?: „Following the original linear TEX86-sea surface temperature (SST) calibration (Schouten et al., 2002), subsequently proposed calibrations include linear (O'Brien et al., 2017) models, including a spatially varying Bayesian approach ('BAYSPAR') (Tierney and Tingley, 2014), and as well reciprocal (Liu et al., 2009) and exponential (Kim et al., 2010) models."

**Author response:** We will change it accordingly.

Line 77-82: Assuming for a moment that the thickness of the early Paleogene mixed layer might have been generally different from today, for example as a result of different intensity of stirring of the upper ocean layers due to invigorated atmosphere dynamics: how would a different water column structure of the early Paleogene tropical Pacific, in particular a different thickness of the mixed layer, impact on the calibration of the target depth, and potentially on results and inferences drawn by the authors in this work? There seems to be evidence that a different thickness of the mixed layer depth cannot be excluded (Quillevere and Norris, 2003; Barnet et al., 2020). Would the impact on peak integrated GDGT source depth be relevant, or are there indications that the effect would be negligible?

**Author response:** We thank the reviewer for pointing this out, the exact thickness of the mixed layer is an uncertainty. We implicitly included this uncertainty in our analyses by taking a conservative depth range of GDGT export and by using two TEX$_{86}$ calibrations (TEX$_{86}^{H}$ for SSTs, and SubT$_{100-250m}$ for SubTs) that together more than encompass the uncertainty of export depth.

We consider that most of the dominant GDGT export was likely from between 50 and 150 meters water depth, particularly because the mixed layer depth, especially in the tropics, is typically shallower than 50 m. GDGTs are typically not found in abundance at depths above the nitracline (e.g., Hurley et al. 2018 (*OG*) because the producing organisms are relatively sensitive to photoinhibition and generally outcompeted (Merbt et al. 2012 (*FEMS*). Two calibrations are used in our work, one at the surface and one between 100 and 250 meters depth, to account for production-depth related uncertainties and to obtain a range of TEX$_{86}$-temperature relationships. This will be further addressed and clarified in the discussion on the depth of the TEX$_{86}$ signal.

Line 90: fix brackets of (Ho and Laepple, 2016)

Line 114: fix brackets of (Kim and O'Neil, 1997)

Line 155: fix brackets of (Lunt et al., 2021):

Line 175: fix „Dashed red dashed line"

**Author response:** All brackets and textual errors mentioned by the reviewer will be fixed.

Line 188: define CIE

**Author response:** Because the Supplementary Text, "CIE" will be introduced before this passage.

Line 196: Is there a specific reason for bold-typesetting the two publications? Fix brackets of (Miller et al., 2020).

**Author response:** This was unintentional and will be fixed.

Line 202: captialize after (c) and fix formatting of d13Corg

Line 211: Fix formatting of (Cramwinckel et al., 2018; Frieling et al., 2019; This Study) (this text is dashed underlined, it is not clear whether this is on purpose)

**Author response:** The mentioned textual and formatting errors will be fixed.

Line 208: Fig. S7 (a-d): Do I correctly interpret that all data points from this site (grey) and from this study (black triangles), are located exactly on the calibration line in subfigures c and d, without any kind of deviation that is apparent to the eye? My apology if I overlooked something obvious, but is there an explanation for this fact? I suggest to give the details of the calibration model, e.g. r-value and fit equation as done for Fig. S1.

**Author response:** For paleo data (grey and black points), SSTs can only be obtained by applying the calibration model to the  TEX$_{86}$ values, hence that they will always fall on the calibration line.

We agree that this may not be intuitive and will clarify this in the caption of Figure S7. Paleo-TEX$_{86}$ data were plotted alongside the present day coretop data, to illustrate the respective TEX$_{86}$ data ranges and the potential issues with extrapolation. Further details on the calibration models (the linear

calibration model from O'Brien et al., 2017 (*ESR*) and the exponential model from Kim et al. 2010 (*GCA*) will be given in the figure caption.

Line 225, Fig. S9: Black and grey dashed lines in a, b, and c look the same to me (insufficient color contrast). Are different colors needed here? If not, just use one color.

**Author response:** The figure and caption will be updated to show only one color.

**References:**

Quillevere, F., and Norris, R. D.: "Ecological development of acarininids (planktonic foraminifera) and hydrographic evolution of Paleocene surface waters", in Causes and consequences of globally warm climates in the early Paleogene, Scott L. Wing, Philip D. Gingerich, Birger Schmitz, Ellen Thomas, 2003.

Barnet, J.S.K., Harper, D. T., LeVay, L. J., Edgar, K. M., Henehan, M. J., Babila, T. L., Ullmann, C. V., Leng, M. J., Kroon, D., Zachos, J. C., and Littler, K.: Coupled evolution of temperature and carbonate chemistry during the Paleocene–Eocene; new trace element records from the low latitude Indian Ocean, Earth and Planetary Science Letters, 545, 116414, https://doi.org/10.1016/j.epsl.2020.116414, 2020.

Lunt, D. J., Bragg, F., Chan, W.-L., Hutchinson, D. K., Ladant, J.-B., Morozova, P., Niezgodzki, I., Steinig, S., Zhang, Z., Zhu, J., Abe-Ouchi, A., Anagnostou, E., de Boer, A. M., Coxall, H. K., Donnadieu, Y., Foster, G., Inglis, G. N., Knorr, G., Langebroek, P. M., Lear, C. H., Lohmann, G., Poulsen, C. J., Sepulchre, P., Tierney, J. E., Valdes, P. J., Volodin, E. M., Dunkley Jones, T., Hollis, C. J., Huber, M., and Otto-Bliesner, B. L.: DeepMIP: model intercomparison of early Eocene climatic optimum (EECO) large-scale climate features and comparison with proxy data, Clim. Past, 17, 203–227, https://doi.org/10.5194/cp-17-203-2021, 2021.

Pithan, F., Mauritsen, T. Arctic amplification dominated by temperature feedbacks in contemporary climate models. Nature Geosci 7, 181–184 (2014). https://doi.org/10.1038/ngeo2071

**References in Author response:**

Locarnini, Mm, Av Mishonov, Ok Baranova, Tp Boyer, Mm Zweng, He Garcia, Jr Reagan, et al. 'World Ocean Atlas 2018, Volume 1: Temperature', 2018. https://archimer.ifremer.fr/doc/00651/76338/.

Cramwinckel, Margot J., Matthew Huber, Ilja J. Kocken, Claudia Agnini, Peter K. Bijl, Steven M. Bohaty, Joost Frieling, et al. 'Synchronous Tropical and Polar Temperature Evolution in the Eocene'. *Nature* 559, no. 7714 (July 2018): 382–86. https://doi.org/10.1038/s41586-018-0272-2.

Gaskell, Daniel E., Matthew Huber, Charlotte L. O'Brien, Gordon N. Inglis, R. Paul Acosta, Christopher J. Poulsen, and Pincelli M. Hull. 'The Latitudinal Temperature Gradient and Its Climate Dependence as Inferred from Foraminiferal $\delta^{18}$O over the Past 95 Million Years'. *Proceedings of the National Academy of Sciences* 119, no. 11 (15 March 2022): e2111332119. https://doi.org/10.1073/pnas.2111332119.

Hollis, Christopher J., Kyle W.R. Taylor, Luke Handley, Richard D. Pancost, Matthew Huber, John B. Creech, Benjamin R. Hines, et al. 'Early Paleogene Temperature History of the Southwest Pacific Ocean: Reconciling Proxies and Models'. *Earth and Planetary Science Letters* 349–350 (October 2012): 53–66. https://doi.org/10.1016/j.epsl.2012.06.024.

Zhang, Yurui, Agatha M. Boer, Daniel J. Lunt, David K. Hutchinson, Phoebe Ross, Tina Flierdt, Philip Sexton, et al. 'Early Eocene Ocean Meridional Overturning Circulation: The Roles of Atmospheric Forcing and Strait Geometry'. *Paleoceanography and Paleoclimatology* 37, no. 3 (March 2022). https://doi.org/10.1029/2021PA004329.

Westerhold, T., U. Röhl, B. Donner, and J. C. Zachos. 'Global Extent of Early Eocene Hyperthermal Events: A New Pacific Benthic Foraminiferal Isotope Record From Shatsky Rise (ODP Site 1209)'. *Paleoceanography and Paleoclimatology* 33, no. 6 (June 2018): 626–42. https://doi.org/10.1029/2017PA003306.

Hurley, Sarah J., Julius S. Lipp, Hilary G. Close, Kai-Uwe Hinrichs, and Ann Pearson. 'Distribution and Export of Isoprenoid Tetraether Lipids in Suspended Particulate Matter from the Water Column of the Western Atlantic Ocean'. *Organic Geochemistry* 116 (February 2018): 90–102. https://doi.org/10.1016/j.orggeochem.2017.11.010.

Merbt, Stephanie N., David A. Stahl, Emilio O. Casamayor, Eugènia Martí, Graeme W. Nicol, and James I. Prosser. 'Differential Photoinhibition of Bacterial and Archaeal Ammonia Oxidation'. *FEMS Microbiology Letters* 327, no. 1 (February 2012): 41–46. https://doi.org/10.1111/j.1574-6968.2011.02457.x.

O'Brien, Charlotte L., Stuart A. Robinson, Richard D. Pancost, Jaap S. Sinninghe Damsté, Stefan Schouten, Daniel J. Lunt, Heiko Alsenz, et al. 'Cretaceous Sea-Surface Temperature Evolution: Constraints from TEX86 and Planktonic Foraminiferal Oxygen Isotopes'. *Earth-Science Reviews* 172 (September 2017): 224–47. https://doi.org/10.1016/j.earscirev.2017.07.012.

Kim, Jung-Hyun, Jaap van der Meer, Stefan Schouten, Peer Helmke, Veronica Willmott, Francesca Sangiorgi, Nalân Koç, Ellen C. Hopmans, and Jaap S. Sinninghe Damsté. 'New Indices and Calibrations Derived from the Distribution of Crenarchaeal Isoprenoid Tetraether Lipids: Implications for Past Sea Surface Temperature Reconstructions'. *Geochimica et Cosmochimica Acta* 74, no. 16 (August 2010): 4639–54. https://doi.org/10.1016/j.gca.2010.05.027.

---

## Author Comment (AC2)

Dear Prof. Guo,

We greatly appreciate the very positive evaluation of our work by Reviewer #2 and thank them for taking another good look at our paper.

Below we reply to each point they raise and indicate how we intend to revise the manuscript.

Sincerely, also on behalf of all co-authors,

Chris Fokkema

Fokkema et al. reported orbitally resolved SSTs and associated data from a tropical Atlantic site. They first confirmed that the Eocene "hyperthermal" events are present in the tropical ocean. Then, these TEX86-based SSTs were used together with benthic foram d18O to constrain Polar Amplification in the ice-free greenhouse world. They identified a persistent PA factor that appears to be robust across different archives, proxy calibrations, and timescales. They concluded that feedback other than ice-albedo, probably involving the carbon cycle, was in play. This is a solid study with a large number of measurements used to provide robust paleoenvironmental reconstructions, while considering potential caveats (such as upwelling at the studied site and different calibrations of TEX86). These results were used to strengthen the series of studies conducted recently by the authors and other folks to constrain PA over different periods of the Cenozoic. I actually reviewed an earlier version of this manuscript for a different journal and don't have many remaining comments. I suggest publication after minor revisions.

**Author response:** We thank the reviewer for their feedback and their positive assessment of the changes made to an earlier version of the manuscript.

The only thing I can think of is to use this opportunity to expand a little bit more on the discussion of feedback mechanisms operating on orbital timescales in the early Cenozoic. This should be feasible given the expertise of the climate dynamics in the author's team. What could be important but currently missing in the model?

**Author response:** While the data-model mismatch does not necessarily imply that the models miss a feedback (i.e., a certain feedback is not included at all), there are multiple feedback mechanisms in the models that still carry large uncertainties. There is a chance that the importance of some of these processes are systematically under- or overestimated, which could result in a (slight) mismatch such as we observe.

We will extend on this in the discussion section, where we will highlight which processes might not be accurately represented in Eocene climate modeling and can cause such as mismatch. This will include cloud feedbacks, which play an important role in polar amplification (Taylor et al., 2013 (*JoC*); Vavrus, 2004 (*JoC*); England and Feldl, 2024 (*JoC*), have large uncertainties in global climate models (Zelinka et al., 2022 (*JGRA*)) and may not be well-simulated in paleoclimate modeling. Several studies have shown that early Eocene high-latitude warmth can be well simulated by changing cloud properties (Sagoo et al., 2013 (*PTRS*); Zhu et al., 2019 (*Sci. Adv.*). In addition, we will discuss ocean heat transport, which influences polar amplification in ice-free climates (England and Feldl, 2024 (*JoC*)). Ocean heat transport in models is sensitive to changes in background conditions, as oceanic gateways, orbital parameters and $CO_2$ concentrations (Huber and Nof., 2006 (*PPP*).

Also, any thought experiment on the potential drivers of these carbon cycle changes on orbital cycles (I know we still haven't really figured out the late Pleistocene G-IG CO2 changes, but what are the potential ways to balance the carbon budget of the Eocene?

**Author response:** The distinct negative $^{13}$C-depleted carbon isotope signature of all hyperthermals of the early Eocene (and in particular the PETM) has led previous workers to the hypothesis of periodic release of $^{13}$C depleted carbon into the ocean-atmosphere system, presumably from an organic reservoir, with proposed reservoirs including methane hydrates and terrestrial organic carbon pools (peatlands, permafrost).

We can conclude that the mechanisms responsible for the eccentricity-forced GMSST variability was likely the same as for the (seemingly superimposed) hyperthermals: the slope of open ocean benthic $\delta^{18}$O versus $\delta^{13}$C during the hyperthermals and during the background variations (including multiple 100-kyr eccentricity scale variations) is equal (Lauretano et al. 2015 (*CotP*)). We will include this in a brief speculative statement in a section at the discussion on carbon cycle feedbacks.

What factors could be more important in the greenhouse world than the Pleistocene)?

**Author response:** The largest difference between the early Eocene situation compared to the Pleistocene is the absence of (sea)ice and snow at the poles, dominantly resulting in a large albedo difference between the Pleistocene G-IG and Eocene greenhouse world. Furthermore, factors like the much warmer background temperatures and higher sea level during the early Eocene presumably affected the importance of surface carbon reservoirs and thereby (orbitally forced) carbon cycle feedback mechanisms. Some of these feedbacks may have been more sensitive in the Pleistocene (e.g., likely more permafrost in the Pleistocene) and others in the Eocene situation (e.g., the sea floor was warmer and therefore the gas hydrate stability zone was smaller, and (high-latitude) peat deposits may have been larger). We will expand on this in the relevant discussion section.

**References in Author response:**

Taylor, Patrick C., Ming Cai, Aixue Hu, Jerry Meehl, Warren Washington, and Guang J. Zhang. 'A Decomposition of Feedback Contributions to Polar Warming Amplification'. *Journal of Climate* 26, no. 18 (15 September 2013): 7023–43. https://doi.org/10.1175/JCLI-D-12-00696.1.

Vavrus, Steve. 'The Impact of Cloud Feedbacks on Arctic Climate under Greenhouse Forcing*'. *Journal of Climate* 17, no. 3 (February 2004): 603–15. https://doi.org/10.1175/1520-0442(2004)017<0603:TIOCFO>2.0.CO;2.

England, Mark R, and Nicole Feldl. 'Robust Polar Amplification in Ice-Free Climates Relies on Ocean Heat Transport and Cloud Radiative Effects'. *JOURNAL OF CLIMATE* 37 (2024).

Zelinka, Mark D., Stephen A. Klein, Yi Qin, and Timothy A. Myers. 'Evaluating Climate Models' Cloud Feedbacks Against Expert Judgment'. *Journal of Geophysical Research: Atmospheres* 127, no. 2 (27 January 2022): e2021JD035198. https://doi.org/10.1029/2021JD035198.

Sagoo, Navjit, Paul Valdes, Rachel Flecker, and Lauren J. Gregoire. 'The Early Eocene Equable Climate Problem: Can Perturbations of Climate Model Parameters Identify Possible Solutions?' *Philosophical Transactions of the Royal Society A: Mathematical, Physical and Engineering Sciences* 371, no. 2001 (28 October 2013): 20130123. https://doi.org/10.1098/rsta.2013.0123.

Zhu, Jiang, Christopher J. Poulsen, and Jessica E. Tierney. 'Simulation of Eocene Extreme Warmth and High Climate Sensitivity through Cloud Feedbacks'. *Science Advances* 5, no. 9 (6 September 2019): eaax1874. https://doi.org/10.1126/sciadv.aax1874.

Huber, Matthew, and Doron Nof. 'The Ocean Circulation in the Southern Hemisphere and Its Climatic Impacts in the Eocene'. *Palaeogeography, Palaeoclimatology, Palaeoecology* 231, no. 1–2 (February 2006): 9–28. https://doi.org/10.1016/j.palaeo.2005.07.037.

Lauretano, V., K. Littler, M. Polling, J. C. Zachos, and L. J. Lourens. 'Frequency, Magnitude and Character of Hyperthermal Events at the Onset of the Early Eocene Climatic Optimum'. *Climate of the Past* 11, no. 10 (7 October 2015): 1313–24. https://doi.org/10.5194/cp-11-1313-2015.

---

## Author Response (AR1)

Dear Prof. Guo,

We thank you for your decision of minor revisions on our manuscript. We have edited the manuscript according to the reviewers comments, and added a few extra textual improvements as well.

Please note that, as suggested by the Reviewer 1, we have transferred the supplementary text to the main manuscript (now sections 2.6 and 2.7), along with supplementary figures S1 and S7 (now Figures 1 and 2, respectively). Furthermore, we added a small map to Figure 1, to show the early Eocene position of the studied location.

All other changes are marked red in the "marked-changes" PDF files. A point-by-point reply to the reviewers comments (including the line numbers of the changes in the "marked-changes" document) is listed below.

Sincerely, also on behalf of all co-authors,

Chris Fokkema

**Comments by RC1:**

**RC1:** *"Overarching comments and appreciation of the study (manuscript text)*

*In their study, Fokkema et al. study polar amplification (PA) of orbital-scale climate variability at a time of the early Cenozoic, when ice sheets were much less wide-spread than they are now, even absent. In comparison to studying Pleistocene climate variability and amplitude of PA, this setting allows separation of the impact of ice-related feedbacks on PA from non-ice-related mechanisms. The authors stress that a big step in their work is the construction of a multi-millenial data set of variability of tropical sea surface temperature. The authors describe the derivation of quantitative inference from cored sediment material at Site 959, discuss and define a calibration to the temperature derivation, and describe an age model that is refined based on previous work. Reliability of climate signals is discussed in detail, as a result of which delta18O is excluded from further analyses.*

*In their supplement, the authors present a concise overview on the calibration of sea water temperature to the geologic archive TEX$^{86}$ that they use, arguing that the use of an exponential calibration is more suited to represent tropical temperatures of warm climates like the Paleogene. I find this argumentation quite important and suggest to move it to the manuscript text."*

**Author response:** We thank the reviewer for this suggestion and have transferred the Supplementary Text to the main text. **[lines 226 – 404]**

**RC1:** "*The work by the authors, that led to this study, is greatly appreciated. For example, this work enables testing the ability of climate models to reproduce PA as recorded in the geologic archive - a valuable opportunity as PA is one of the relevant climate system metrics for understanding and projecting future climate at much warmer than modern high latitudes. This work may hence extend our model validation from the very short modern observational period towards past (future- analogue) climates. The utility of the presented data towards exactly this purpose is demonstrated by the authors at the example of PA in an ice-free EECO climate. Based on a comparison of their sea surface temperature reconstruction with DeepMIP model simulations presented by Lunt et al. (2021), the authors infer that models agree well with inferences from the geologic record, while noting that PA-*

*causing processes and mechanisms, aside from those related to ice, may be underestimated in the models.*

*The authors present various insights that are very valuable. Their work supports linkages between climate variability of an early Cenozoic hothouse world with dynamics on glacial-interglacial cycles, posing the question to which extent carbon-cycle feedbacks played a role in Pleistocene Milancovitch cycles. Furthermore, the work illustrates the magnitude of current climate change in the context of Cenozoic climate history. I find the statement „modern GMSST warming of ~1 °C is already in the range of the early Eocene hyperthermal events" particularly remarkable.*

*In my opinion the manuscript is very well, and carefully, written. I have located several minor issues and provide some comments at locations where I, as an interested reader, would like to have a bit more insight regarding specific aspects of the work. One addition to the discussion / conclusion outlook could be to propose testing whether the findings of this study will be reproduced in analyses from other cores and from other core locations at a similar latitude. I have no reason to doubt the assumptions made by the authors, regarding both spatial representativeness of the reconstructed signal of sea surface temperature variability and depth of the water column to which a reconstructed temperature signal is attributed; and, as the authors note, their assumptions are supported by auxiliary evidence, as for example by climate modelling. Nevertheless, further testing of these assumptions based on material from other cores is, at least in my opinion, worthwhile. Spending one or two related sentences in the discussion, and/or in an outlook section, would in my opinion provide a meaningful conclusion to the manuscript. "*

**Author response:** We thank the reviewer for this recommendation. We have included the following statement in the discussion:

*"Finally, although the climate models show that SST variability at the location of Site 959 should be an adequate tracer for SST variability in the complete tropical band (Fig. 1b), we stress the need for additional high resolution, early Eocene SST reconstructions. Specifically, new records from low latitudes would be of great value for validating and extending our record, and reconstructions from high latitudes will provide an optimal comparison for assessing PA."* [**lines 558–562**].

**RC1:** "*I note that I am not a climate scientist conducting analyses of the sedimentary records similar to those analyses described by the authors. Hence, my knowledge in details of sample preparation procedures and analysis methodology is not very deep. If the editors have any doubts regarding the validity of the analysis methodology, then I kindly ask to refer to an expert in that field for a second opinion.*

*In summary, I support publication of the study in Climate of the Past subsequent to addressing or rebuting comments."*

**Author response:** We thank the reviewer for their appreciation of our work.

**RC1:** "*Specific comments (main text)*

Terminology regarding geologic timescales: I noted that in their supplement the authors refer several times to Paleogene or early Paleogene, while the term barely appears in the main text. When the authors speak of the Early Paleogene, do they refer to the (early) Eocene that is often referred to in the main text? I do not think that the link is always clear - in particular, to my understanding, the early Paleogene would likely rather refer to the Paleocene than to the Eocene? I may be wrong here, but clarifying the text where necessary may be helpful for readers. "

**Author response:** We realize this might be confusing for members of our community that are not familiar to the details of stratigraphy and stratigraphical nomenclature. For clarity, "Paleogene" is changed to "Eocene" to remain consistent in the terminolgy.

**RC1:** "*Line 24: add comma after „orbital"*

*Line 44: add comma after „feedbacks to PA" "*

**Author response:** Commas are added.

**RC1:** "*Lines 40-49: re origin of PA in climate models: I suggest to also refer to / comment on the latitude-dependency of the fraction of outgoing radiation (Pithan and Mauritsen, 2014) via temperature feedbacks, as these have been found to dominate the mechanism for PA in CMIP5 models. "*

**Author response:** This is done.

**RC1:** "*Line 102: Meaning of the text „no ice and continental configuration" remains unclear. Do the authors aim at the degree of detail of paleogeography considered in the Eocene simulations? If so, should this text rather read „adaptation of model geography to reconstructed continental configuration and absence of major ice sheets" or similar?"*

**Author response:** This is clarified to *"(i.e., early Eocene paleogeography without ice sheets)"* [**Line 104**].

**RC1:** "*Line 154: add „for" after 0.07‰*

*Line 156: remove space between 13 and %*

*Line 158: were -> was*

*Line 164: remove spaces between values and ‰ and % signs, respectively*

*Line 219: I am not sure whether the term „neutralize" is correct here. Tap water is not neutralizing afaik, it is rather diluting - does'nt one need to add a base in order to neutralize an acid? "*

**Author response:** Above textual errors are fixed, and "neutralized"is changed to "diluted"

**RC1:** *Line 252: change to „a climate signal" or „as a climate signal"*

**Author response:** This is done

**RC1:** *Line 261: fix the format of the DOI? (remove the space and/or make the doi a hyperlink?)*

**Author response:** this is fixed

**RC1:** *Line 264: plots of CENOGRID are at least to me confusing due to the same / very similar color being chosen for both benthic d13C and benthic d18O. Based on the alignment of data and y-axis I can guess which branch represents which isotopic ratio, but the color coding is not helpful here. The „bracket"-like signal on the right, near 56 Myr, is unclear to me - please explain if relevant.*

**Author response:** The layout of this figure is now changed, so that the distinction between CENOGRID and Site 959, and the different isotopes are more clear. The "bracket-like" feature during the PETM (56 Myr) is fixed by flipping the $d^{13}C$ axis, so all excursions phase upwards. (**Fig. 3**)

**RC1:** *Line 272: Maybe provide the modern temperatures in the region as a reference for the 35.2°C of Eocene SST towards providing a rough estimate of climatic difference wrt. to today?*

**Author response:** We added the modern regional SSTs "*present-day annual average SST is 27.7 ℃ (Locarnini et al., 2018)*" [**Line 454**]

**RC1:** *Page 10, Fig. 2: Clarify the meaning of „E-08" of the susceptibility record - shall this be $10{-}8$? Refer to my comment in Fig. 1 regarding colors of CENOGRID records. I assume the reference (c) in the figure caption should be moved from the end of the sentence to before „Calcareous nannofossil zones"?*

**Author response:** "E-08" is changed to "$\times\,10^{-8}$". "*(c)*" is moved to the front of the respective sentence. Colors of the CENOGRID are changed, as for Fig. 3. [**Fig. 4**]

**RC1:** *Line 282: „The record shows" or „The records show"*

**Author response:** "The records shows" is adjusted to "The $TEX_{86}$ record shows" [**Line 464**]

**RC1:** *Line 286: Refer to my suggestion to move much of the information from the supplement into the manuscript text. Yet, even if this is not done, my feeling is that a bit more information regarding the model simulations should be given here than just a reference to the supplement.*

**Author response:** We have moved the relevant parts of the supplement to the main text, including the information about the model simulations [**Lines 399 – 404**].

**RC1:** *Line 288: „of both records": Maybe once more explicitly state which records you refer to, for clarity. Same for „the dataset".*

**Author response:** This is done.

**RC1:** *Line 289: Should SD be in singular here? If plural, maybe there is a problem with the formulation.*

**Author response:** "SDs" is changed to "SD"

**RC1:** *Line 292: Indeed, the calibration uncertainty / analytical errors are small regarding the absolute values, but they are large in comparison to the reconstructed SD?*

**Author response:** We have slightly reworded this part optimize clarity in such a way that the shortcomings of simply comparing SDs, in particular related to the analytical errors, are better emphasized.

" *The analytical errors (~0.2 ℃ for SST (see Methods) and ~0.4 ℃ for $\delta^{18}O$-BWT (see section 2.7) are relatively large compared to the SDs of the complete records. However, the analytical errors are too small to explain the complete offset in variability of the two records. This implies that the obtained value is a crude approximation of PA..*" [**Lines 473–476**]

**RC1:** *Line 325: Fig. 3 suggests that especially CIE J and K coincide with a state of relative warmth rather than a state or warming, that may even include a subsequent cooling, but not necessarily only a phase of transient warming - am I right with this observation?*

**Author response:** We have clarified this by describing the temperature effects during the events here as "warmer intervals" . [**Line 508**]

**RC1:** *Line 365: Do the authors refer to the whole tropical band or only to the Northern Hemisphere part?*

**Author response:** We here refer to the complete low-latitude band, and this is now clarified by changing "<30º" to "30º S – 30º N". [**Line 549**]

**RC1:** *Furthermore, regarding the statement „the dominant source of Eocene bottom waters in these simulations" - is this an inference that authors make based on their own analyses, or do the authors refer to results by Lunt et al. (2021), or maybe even to results from authors who contributed model simulations towards the model intercomparison by Lunt et al. (2021)? If the result is not derived by the authors themselves, then I assume citing the relevant publication(s) in this specific context makes sense.*

**Author response:** For clarity, we have added references (e.g., Huck et al. 2017, Hollis et al. 2012, Zhang et al. 2022) to this statement. [**line 550**]

**RC1:** *Figure 5, caption: fix bracket of (Herbert et al., 2010)*

**Author response:** The bracket is fixed.

**RC1:** *Lines 416-418: Simplify reading by adding some commas: „[...] carbon cycle feedbacks, that do not involve ice, snow and frost-related processes, were only inherent to past greenhouse climate, [...]"*

**Author response:** Commas are added.

**RC1:** *Lines 424-426: please check the sentence „[...] we conclude that early Eocene PA is not impacted by non-ice feedback mechanisms that act on $10^4$-year timescales or longer."*

**Author response:** This sentence has been changed to:

*"we conclude that early Eocene PA is dominated by non-ice feedback mechanisms that act on $10^4$-year timescales or shorter."* [**lines 646–648**].

**RC1:**

*Specific comments (Supplementary material):*

*In my opinion the supplementary material present information that is key to a full appreciation of the work presented by the authors. In my opinion the text on pages 2-6 is actually very relevant for a deeper understanding of the work. In particular section 1.1, but also the other sections, would in my humble opinion fit well into the manuscript text. If there is no good reason to put this text into a supplement, I would suggest to move it to the manuscript. I think the description of employed model output (lines 154-159) should really be presented in a data section to make the link between proxy data analysis and climate modeling more clear in the manuscript.*

**Author response:** We have moved all the supplementary text to the main text, including the section on TEX$_{86}$ and model outputs. "early Paleogene" has been changed to "(early) Eocene". [**lines 226 – 404**]

**RC1:** *In this context please highlight that the simulations by Lunt et al. (2021) represent climate states of the early Eocene climate optimum (EECO, ~ 50 million years ago). Is there any need to „extrapolate" results derived from these simulations to different periods described in the manuscript, in particularly an early Paleogene climate (I am not quite sure about the definition of early Paleogene)?*

**Author response:** Our dataset covers the onset of the EECO (*ca.* 53 – 49 Ma, e.g. Westerhold et al. 2018, *P&P*), and match the target of DeepMIP simulations that we compare to (i.e., the boundary conditions regarding ice sheets and continental configuration).

**RC1:** *Line 22-24: I think the text is potentially ambiguous, should it read as follows?: „Following the original linear TEX86-sea surface temperature (SST) calibration (Schouten et al., 2002), subsequently proposed calibrations include linear (O'Brien et al., 2017) models, including a spatially varying Bayesian approach ('BAYSPAR') (Tierney and Tingley, 2014), and as well reciprocal (Liu et al., 2009) and exponential (Kim et al., 2010) models.“*

**Author response:** We have changed this as suggested [**lines 236–238**]

**RC1:** *Line 77-82: Assuming for a moment that the thickness of the early Paleogene mixed layer might have been generally different from today, for example as a result of different intensity of stirring of the upper ocean layers due to invigorated atmosphere dynamics: how would a different water column structure of the early Paleogene tropical Pacific, in particular a different thickness of the mixed layer, impact on the calibration of the target depth, and potentially on results and inferences drawn by the authors in this work? There seems to be evidence that a different thickness of the mixed layer depth cannot be excluded (Quillevere and Norris, 2003; Barnet et al., 2020). Would the impact on peak integrated GDGT source depth be relevant, or are there indications that the effect would be negligible?*

**Author response:** We thank the reviewer for pointing this out, the exact thickness of the mixed layer is an uncertainty. We implicitly included this uncertainty in our analyses by taking a conservative depth range of GDGT export and by using two $TEX_{86}$ calibrations ($TEX_{86}^{H}$ for SSTs, and $SubT_{100-250m}$ for SubTs) that together more than encompass the uncertainty of export depth.

We consider that most of the dominant GDGT export was likely from between 50 and 150 meters water depth, particularly because the mixed layer depth, especially in the tropics, is typically shallower than 50 m. GDGTs are typically not found in abundance at depths above the nitracline (e.g., Hurley et al. 2018 (*OG*) because the producing organisms are relatively sensitive to photoinhibition and generally outcompeted (Merbt et al. 2012 (*FEMS*). [**Lines 300–301**]

Two calibrations are used in our work, one at the surface and one between 100 and 250 meters depth, to account for production-depth related uncertainties and to obtain a range of $TEX_{86}$-temperature relationships. This is further emphasized and clarified in the discussion. [**Lines 338–339**]

**RC1:** *Line 90: fix brackets of (Ho and Laepple, 2016)*

*Line 114: fix brackets of (Kim and O'Neil, 1997)*

*Line 155: fix brackets of (Lunt et al., 2021):*

*Line 175: fix „Dashed red dashed line“*

**Author response:** All brackets and textual errors mentioned by the reviewer are fixed.

**RC1:** *Line 188: define CIE*

**Author response:** Because the Supplementary Text is moved to the main text, "CIE" is now introduced before this passage. [**Line 81**]

**RC1 :**_Line 196: Is there a specific reason for bold-typesetting the two publications? Fix brackets of (Miller et al., 2020)._

**Author response:** This was unintentional and is fixed.

**RC1:** _Line 202: captialize after (c) and fix formatting of d13Corg_

_Line 211: Fix formatting of (Cramwinckel et al., 2018; Frieling et al., 2019; This Study) (this text is dashed underlined, it is not clear whether this is on purpose)_

**Author response:** The mentioned textual and formatting errors are fixed.

**RC1:** _Line 208: Fig. S7 (a-d): Do I correctly interpret that all data points from this site (grey) and from this study (black triangles), are located exactly on the calibration line in subfigures c and d, without any kind of deviation that is apparent to the eye? My apology if I overlooked something obvious, but is there an explanation for this fact? I suggest to give the details of the calibration model, e.g. r-value and fit equation as done for Fig. S1._

**Author response:** For paleo data (grey and black points), SSTs can only be obtained by applying the calibration model to the $TEX_{86}$ values, hence that they will always fall on the calibration line.

We agree that this may not be intuitive and clariefied this in the caption of (now) Figure 2. **[lines 289–290]**.

Paleo-$TEX_{86}$ data were plotted alongside the present day coretop data, to illustrate the respective $TEX_{86}$ data ranges and the potential issues with extrapolation. Further details on the calibration models (the linear calibration model from O'Brien et al., 2017 (_ESR_) and the exponential model from Kim et al. 2010 (_GCA_) aree given in the figure caption. **[lines 287–288]**.

**RC1:** _"Line 225, Fig. S9: Black and grey dashed lines in a, b, and c look the same to me (insufficient color contrast). Are different colors needed here? If not, just use one color. "_

**Author response:** The figure and caption are updated to show only one color.

**Comments by RC2:**

**RC2:** _"Fokkema et al. reported orbitally resolved SSTs and associated data from a tropical Atlantic site. They first confirmed that the Eocene "hyperthermal" events are present in the tropical ocean. Then, these TEX86-based SSTs were used together with benthic foram d18O to constrain Polar Amplification in the ice-free greenhouse world. They identified a persistent PA factor that appears to be robust across different archives, proxy calibrations, and timescales. They concluded that feedback other than ice-albedo, probably involving the carbon cycle, was in play. This is a solid study with a large number of measurements used to provide robust paleoenvironmental reconstructions, while considering potential caveats (such as upwelling at the studied site and different calibrations of TEX86). These results were used to strengthen the series of studies conducted recently by the authors and other folks to constrain PA over different periods of the Cenozoic. I actually reviewed an earlier version of this manuscript for a different journal and don't have many remaining comments. I suggest publication after minor revisions."_

**Author response:** We thank the reviewer for their feedback and their positive assessment of the changes made to an earlier version of the manuscript.

**RC2: "**_The only thing I can think of is to use this opportunity to expand a little bit more on the discussion of feedback mechanisms operating on orbital timescales in the early Cenozoic. This should be feasible given the expertise of the climate dynamics in the author's team. What could be important but currently missing in the model?_"

**Author response:** While the data-model mismatch does not necessarily imply that the models miss a feedback (i.e., a certain feedback is not included at all), there are multiple feedback mechanisms in the models that still carry large uncertainties. There is a chance that the importance of some of these processes are systematically under- or overestimated, which could result in a (slight) mismatch such as we observe.

We have extended on this subject in the discussion:

_"As cloud feedbacks present one of the largest uncertainties in climate models (Zelinka et al., 2022), and play an important role in ice-free PA (Vavrus, 2004; Taylor et al., 2013a; England and Feldl, 2024), it is plausible that their effect is underestimated in the models. Indeed, several studies have shown that early Eocene high-latitude warmth can be better simulated by changing model cloud properties (Sagoo et al., 2013; Zhu et al., 2019). In addition, ocean heat transport, which is likely important for PA in ice-free climates (England and Feldl, 2024), is sensitive to changes in oceanic gateways or other factors such as orbital parameters and CO2 concentrations (Huber and Nof, 2006), which might not be accurately represented in Eocene climate modeling (Zhang et al. 2022)."_ [**lines 569–575**]

**RC2:** "_Also, any thought experiment on the potential drivers of these carbon cycle changes on orbital cycles (I know we still haven't really figured out the late Pleistocene G-IG CO2 changes, but what are the potential ways to balance the carbon budget of the Eocene?_ "

**Author response:** The distinct negative [13]C-depleted carbon isotope signature of all hyperthermals of the early Eocene (and in particular the PETM) has led previous workers to the hypothesis of periodic release of [13]C depleted carbon into the ocean-atmosphere system, presumably from an organic reservoir, with proposed reservoirs including methane hydrates and terrestrial organic carbon pools (peatlands, permafrost).

We have included a brief speculative statement on the possible carbon sources in the discussion:

_"The eccentricity-forced GMSST variations during the early Eocene implies, as suggested by previous work (Vervoort et al., 2021), a sensitive (organic) carbon cycle feedback mechanism at play. It has previously been demonstrated that the benthic $\delta^{13}C$-$\delta^{18}O$ slope is equal during the hyperthermals and the background variations (Lauretano et al., 2015). This strongly suggests that the same carbon reservoirs and feedback mechanisms were responsible for the 405-kyr eccentricity-paced hyperthermals and the ~100-kyr eccentricity-paced climate variability. The proposed carbon capacitors responsible for hyperthermal variability include soils and peats (Kurtz et al., 2003), permafrost (DeConto et al., 2012) and methane hydrates (Dickens et al., 1997; Komar et al., 2013), all characterized by strong negative $\delta^{13}C$ signatures. Importantly, as the carbon release and warming is clearly of much greater magnitude during some of the larger, 405-kyr, eccentricity maxima, we can conclude that the characteristics of the responsible carbon cycle feedback must include that it can both scale with orbital forcing, and can accelerate after reaching a threshold or tipping point (Setty et al., 2023). Finally,_

*although the exact mechanisms are uncertain, we note that the volume of $CO_2$ released via this (combination of) feedback mechanism(s) must be large.* **[lines 602–613]**

**RC2:** "*What factors could be more important in the greenhouse world than the Pleistocene)?*"

**Author response:** We have extended the discussion on differene between Pleistocene and Eocene situation, and which factors could be more important:

*"The different background conditions of the early Eocene compared to the Pleistocene, which, apart from significant differences in ice-related surface albedo, includes higher background temperatures, higher sea level and a different paleogeography, could have caused certain carbon cycle feedback mechanisms to act differently than in the Pleistocene. For example, it can be assumed that the warm Eocene climate would have largely restricted the area of permafrost in the Antarctic interior that experienced relatively warm and wet summers (Baatsen et al., 2024). Intriguingly, the carbon storage in permafrost may have been replaced by extensive peat deposits with, presumably, a similar carbon cycle impact. Moreover, the higher seafloor temperatures would greatly reduce the potential volume of methane hydrates (Dickens, 2001b), although a long-term warming since the Paleocene might have put the methane hydrates closer to a critical threshold (Zachos et al., 2001)."* [**Lines 629–637**]